# Attenuating Effects of Dieckol on Hypertensive Nephropathy in Spontaneously Hypertensive Rats

**DOI:** 10.3390/ijms22084230

**Published:** 2021-04-19

**Authors:** Myeongjoo Son, Seyeon Oh, Junwon Choi, Ji Tae Jang, Kuk Hui Son, Kyunghee Byun

**Affiliations:** 1Department of Anatomy & Cell Biology, Gachon University College of Medicine, Incheon 21936, Korea; mjson@gachon.ac.kr; 2Functional Cellular Networks Laboratory, Department of Medicine, College of Medicine, Graduate School and Lee Gil Ya Cancer and Diabetes Institute, Gachon University, Incheon 21999, Korea; seyeon8965@gmail.com (S.O.); choijw88@gc.gachon.ac.kr (J.C.); 3Aqua Green Technology Co., Ltd., Smart Bldg., Jeju Science Park, Cheomdan-ro, Jeju 63243, Korea; whiteyasi@gmail.com; 4Department of Thoracic and Cardiovascular Surgery, Gachon University Gil Medical Center, Gachon University, Incheon 21565, Korea

**Keywords:** epithelial-to-mesenchymal transition, E. cava extracts, dieckol, spontaneously hypertensive rats, renal fibrosis, angiotensin II

## Abstract

Hypertension induces renal fibrosis or tubular interstitial fibrosis, which eventually results in end-stage renal disease. Epithelial-to-mesenchymal transition (EMT) is one of the underlying mechanisms of renal fibrosis. Though previous studies showed that Ecklonia cava extracts (ECE) and dieckol (DK) had inhibitory action on angiotensin (Ang) I-converting enzyme, which converts Ang I to Ang II. It is known that Ang II is involved in renal fibrosis; however, it was not evaluated whether ECE or DK attenuated hypertensive nephropathy by decreasing EMT. In this study, the effect of ECE and DK on decreasing Ang II and its down signal pathway of angiotensin type 1 receptor (AT1R)/*TGFβ*/*SMAD*, which is related with the EMT and restoring renal function in spontaneously hypertensive rats (SHRs), was investigated. Either ECE or DK significantly decreased the serum level of Ang II in the SHRs. Moreover, the renal expression of AT1R/*TGFβ*/*SMAD* was decreased by the administration of either ECE or DK. The mesenchymal cell markers in the kidney of SHRs was significantly decreased by ECE or DK. The fibrotic tissue of the kidney of SHRs was also significantly decreased by ECE or DK. The ratio of urine albumin/creatinine of SHRs was significantly decreased by ECE or DK. Overall, the results of this study indicate that ECE and DK decreased the serum levels of Ang II and expression of AT1R/*TGFβ*/*SMAD*, and then decreased the EMT and renal fibrosis in SHRs. Furthermore, the decrease in EMT and renal fibrosis could lead to the restoration of renal function. It seems that ECE or DK could be beneficial for decreasing hypertensive nephropathy by decreasing EMT and renal fibrosis.

## 1. Introduction

The kidney is a major organ that is affected by hypertensive target organ damage: chronic kidney disease commonly occurs in around 16% of hypertensive patients [1].

The pathological features of hypertensive nephropathy include inflammation, glomerular sclerosis, tubular atrophy, and interstitial fibrosis [2]. The fibrotic tissue replaces the normal functional kidney tissue, which leads to renal failure [3,4]. Thus, renal fibrosis or tubular interstitial fibrosis is the main pathological lesion of hypertensive nephropathy, which induces end-stage renal disease (ESRD) [5]. Renal fibrosis shows the following characteristics: increased massive extracellular matrix (ECM) production, increased recruitment of fibroblasts to tissue injury sites, and increased phenotype changes from fibroblasts to α-smooth muscle actin (α-SMA)-expressing myofibroblasts [6]. The increased α-SMA-expressing myofibroblasts lead to the unnecessarily excessive deposition of collagen and enhance the dysregulation of matrix metalloproteinases, which destroys the basement membrane and further accelerates fibrosis [6].

Recently, many studies have shown that epithelial-to-mesenchymal transition (EMT) is one of the underlying mechanisms of renal fibrosis [7,8]. By undergoing EMT, the epithelial cell gradually loses epithelial markers, such as E-cadherin, which is the main element of cell-to-cell junctions, and gains markers of mesenchymal phenotype, such as α-SMA [9]. By losing the cell junctions, the cells that underwent the EMT process easily move toward the interstitial space [9]. Moreover, the epithelial cells that acquired the properties of myofibroblasts express α-SMA and synthesize ECM proteins, such as collagen, eventually leading to tubular interstitial fibrosis (TIF) [9].

Angiotensin II (Ang II) acts as the main player in hypertension-induced fibrosis. As the main activator of the renin–angiotensin–aldosterone system (RAAS), Ang II causes severe vascular, glomerular, and tubulointerstitial injuries, which are accompanied by the increase in transforming growth factor-beta (TGFβ) via the angiotensin type 1 receptor (AT1R) [10]. The levels of Ang II and its receptor in the kidneys of the spontaneously hypertensive rats (SHRs) were higher than those of Wistar Kyoto (WKY) rats [11]. It is known that Ang II induces EMT via TGFβ-dependent signaling pathways [12]. The Ang II treatment for NRK52E cells (normal rat tubular epithelial cell lines) induces the expression of the TGFβ1/SMAD signaling pathway, which eventually increases the proapoptotic and fibrotic proteins [13]. When Ang II binds with AT1R, the SMAD3 signal pathway is activated and induces EMT in NRK52E cell lines [12].

Polyphenols from marine algae have been reported to function as Ang I-converting enzyme (ACE) inhibitors [14,15,16,17]. ACE is involved in the conversion of Ang I to Ang II, and increases Ang II as a potent vasoconstrictor, leading to hypertension [18]. Several phlorotannins, such as phlorofucofuroeckol A, dieckol (DK), and eckol, which are present in extracts from Ecklonia cava or Ecklonia stolonifera, show ACE inhibiting activities; thus, these polyphenols were expected to decrease blood pressure (BP) [15,19]. Even though E. cava extracts (ECE) and DK show ACE inhibiting activities, no report has investigated whether ECE or dieckol is involved in the EMT of the kidney or renal fibrosis that is induced by hypertension. Thus, the effect of ECE and DK on decreasing Ang II and its down signal pathway of AT1R/TGFβ/SMAD2/3, which is related with the EMT of the kidney and restoring renal function in SHRs, was investigated in this study.

## 2. Results

### 2.1. ECE and DK Reduced Systolic BP and Serum Level of Ang II in SHRs

The systolic BP of the SHRs was significantly higher than that of the WKY rats and was significantly decreased by the administration of either ECE or DK. The decreasing effects of 50, 100, and 150 mg/kg/day of ECE and DK administration were not significantly different (Figure 1a). The diastolic and mean BPs of the SHRs were significantly higher than those of the WKY rats. The administration of either ECE or DK did not significantly decrease the diastolic and mean BP (Figure 1b,c).

The serum level of Ang II in the SHRs was significantly higher than that of the WKY rats and was significantly decreased by the administration of either ECE or DK. Moreover, the most prominent decreasing effect was observed in the group treated with 150 mg/kg/day of ECE (Figure 1d).

### 2.2. ECE and DK Attenuated the Expression of AT1R in the Kidney of SHRs

The expression of AT1R was evaluated through qRT-PCR and staining. The mRNA expression of AT1R in the medulla and cortex of the kidney was significantly higher in the SHRs than that of the WKY rats. Such expression was decreased by the administration of either ECE or DK (Figure 2a). In the medulla, the decreasing effect was most prominent in the group treated with DK. In the cortex, 150 mg/kg/day of ECE had the most prominent decreasing effect. The expression level of AT1R in the cortex and medulla of the SHRs, which was evaluated through staining, was significantly increased and was significantly decreased by the administration of either ECE or DK. Moreover, the most prominent decreasing effect was observed in the group treated with DK (Figure 2b,c).

To validate ECE and DK attenuation of AT1R expression in kidney tubules, a mouse proximal tubule cell line (TCMK-1) was activated by angiotensin II in an in vitro model [20]. The expression level of AT1R protein in angiotensin II-treated TCMK-1 cells with ECE (5, 25, 50 ug/mL), DK or inhibitor was decreased compared to only angiotensin II treatment (Figure 2d).

### 2.3. ECE and DK Reduced the Expression of TGFβ, SMAD2/3, and Snail2 in the Kidney of SHRs

The expressions of TGFβ in the medulla and cortex of the SHRs were significantly higher than those of the WKY rats, which were decreased by the administration of ECE or DK. In the medulla, the decreasing effect was most prominent in the groups treated with 150 mg/kg/day of ECE and DK. In the cortex, the decreasing effect was most prominent in the group treated with DK (Figure 3a). The expressions of SMAD2 in the medulla and cortex of the SHRs were higher than those of the WKY rats, which were significantly decreased by the administration of ECE or DK. The decreasing effect in the medulla was most prominent in the groups treated with 150 mg/kg/day of ECE or DK. In the cortex, the decreasing effects of 50, 100, and 150 mg/kg/day of ECE or DK were not significantly different (Figure 3b). The expression of SMAD3 in the medulla and cortex of the SHRs was significantly higher than that of the WKY rats, and was significantly decreased by the administration of ECE or DK. The most prominent decreasing effect was observed in the group treated with 150 mg/kg/day of ECE (Figure 3c). The expression of Snail2 in the medulla and cortex of the SHRs was significantly increased by the administration of ECE or DK. The most prominent decreasing effect was observed in the group treated with DK (Figure 3d). In the in vitro model, the expression level of TGFβ and pSMAD2/3 protein in angiotensin II-treated TCMK-1 cells with ECE (5, 25, 50 ug/mL), DK or inhibitor was decreased compared to the only angiotensin II treatment (Figure 3e,f).

### 2.4. ECE and DK Reduced the EMT in the Kidney of the SHRs

The expression of the mesenchymal cell marker, such as vimentin and α-SMA, of the SHRs was significantly higher than that of the WKY rats and was significantly decreased by the administration of either ECE or DK (Figure 4). The decreasing effect on the expression of vimentin in the cortex of the SHRs was most prominent in the group treated with DK, whereas the decreasing effect on the expression of vimentin in the medullas of the SHRs was most prominent in the group treated with 150 mg/kg/day of ECE. Moreover, the decreasing effect on the expression of vimentin in the cortex of the SHRs was most prominent in the group treated with DK (Figure 4a,b). The expression of α-SMA in the cortex and medulla of the SHRs was most significantly decreased in the group treated with 150 mg/kg/day of ECE (Figure 4c,d). The expression of the epithelial cell marker, such as E-cadherin, in the cortex and medulla of the SHRs was significantly lower than that of the WKY rats (Appendix A) and was significantly increased by the administration of either ECE or DK.

### 2.5. ECE and DK Reduced the Renal Fibrosis and Glomerular Sclerosis in the Kidney of SHRs

The fibrosis area in the cortex and medulla of the kidney of the SHRs was significantly higher than that of the WKY rats (Figure 5a,b) and was significantly decreased by the administration of either ECE or DK. In the cortex, the most prominent decreasing effect was observed in the group treated with DK, whereas in the medulla, the most prominent decreasing effect was observed in the group treated with 150 mg/kg/day of ECE. The GSI of the SHRs was significantly higher than that of the WKY rats (Figure 5c,d) and was significantly decreased by the administration of ECE or DK. Moreover, the most prominent decreasing effect was observed in the group treated with DK.

### 2.6. ECE and DK Attenuated Renal Function Aggravation in SHRs

The amount of water intake within 24 h was not significantly different among all groups (Figure 6a). The urine volume of the SHRs within 24 h was significantly lower than that of the WKY rats (Figure 6b). It was significantly increased by the administration of 100 and 150 mg/kg/day of ECE and DK. The urine sodium/potassium (Na/K) ratio was not significantly different among all groups (Figure 6c). The albumin level in the urine of the SHRs was significantly higher than that of the WKY rats and was significantly decreased by the administration of ECE or DK. The decreasing effects of 50, 100, and 150 mg/kg/day of ECE and DK were not significantly different (Figure 6d). The ratio of urine albumin/creatinine of the SHRs was significantly higher than that of the WKY rats and was decreased by the administration of ECE or DK. The decreasing effects of 50 and 100 mg/kg/day of ECE and DK were not significantly different (Figure 6e).

## 3. Discussion

Hypertension is the second leading etiology of ESRD after diabetes [21]. It is hard to predict the severity of hypertensive renal fibrosis with BP, since renal fibrosis could severely progress even when the patient’s BP is not extremely high [21]. Although antihypertensive treatments with ACE inhibitors, such as angiotensin receptor blockers, renin inhibitors, and aldosterone antagonists, could reduce the severity of hypertensive kidney disease, they are not enough to prevent the progression of hypertensive nephropathy [22]. Even though the recommended target BP is achieved at below 130/80 mmHg, the treatment strategy for decreasing BP cannot delay the progression of hypertensive nephropathy [23]. Although hypertensive nephropathy is typically described as nephroangiosclerosis and glomerular hyalinosis [24,25], recently, it was revealed that the interstitium of the kidney, which is involved in TIF, is related to disease progression, as well as the glomerular and vascular compartments [26,27]. Since TIF is a main pathophysiology of hypertensive nephropathy, it is essential to the development of new agents directed to modulate TIF to prevent the progression of hypertensive nephropathy. In recent years, EMT has been known as a crucial player in renal fibrosis [28,29].

Ang II induces vasoconstriction and consequently induces hypertension [30]. In hypertension, RAAS is activated systemically, and the upregulation of RAAS is accompanied in several organs, such as the kidney [31,32]. The hyperactivation of intrarenal RAAS is an important mechanism of hypertension and chronic kidney disease [31,32]. Both TGFβ1 and Ang II induce the activation of the intrarenal RAAS and enhance the expressions of angiotensinogen, renin, ACE, and AT1R, which induce EMT [33]. In our study, the systolic BP of the SHRs was significantly decreased by the administration of either ECE or DK. The serum level of Ang II of the SHRs was significantly decreased by the administration of either ECE or DK. The expression of AT1R in the cortex and medulla of the SHRs was significantly decreased by the administration of either ECE or DK.

The upregulation of RAAS, such as the increased expression of AT1R, activates the down signaling pathway of TGFβ1 [34]. Activated by RAAS, TGFβ1 induces EMT via the SMAD-dependent and SMAD-independent pathways [33]. As the SMAD-dependent signaling pathway, TGFβ1 signaling is transduced via TGFβRII and TGFβRI and leads to the activation of SMAD2/3. [10]. SMAD4 binds to SMAD2/3 and promotes the translocation of the SMAD complex into the nucleus [10]. Translocated SMADs upregulate the expressions of EMT genes, such as Snail, Twist, and ZEB [34]. In our study, the expression of AT1R was significantly higher in the cortex and medulla of the SHRs than that of the WKY rats, and was decreased by the administration of either ECE or DK. The expression of TGFβ and SMAD2/3 in the cortex and medulla of the SHRs was decreased by the administration of either ECE or DK.

The EMT is characterized by the loss of an epithelial marker, such as E-cadherin, and the attainment of mesenchymal markers, such as α-SMA, vimentin, and fibronectin [35]. E-cadherin is the most expressed cadherin in the epithelial cells and is, thus, frequently used as an epithelial marker [36]. α-SMA is a phenotypic marker of myofibroblast cells, and its expression is a feature of the advanced stages of EMT [37]. α-SMA-expressing myofibroblasts, which undergo EMT, induce the synthesis of excessive ECM and abnormal ECM remodeling and renal fibrosis [37].

Hypertensive nephropathy induces glomerulosclerosis and interstitial fibrosis, which induces the decrease in renal function and significant proteinuria [6]. In our study, the expression of vimentin and α-SMA in the medulla and cortex of the SHRs was increased and was decreased by the administration of either ECE or DK. The amount of fibrosis area in the medulla and cortex of the SHRs was increased, and it was decreased by the administration of either ECE or DK. The index of glomerular sclerosis of the SHRs was increased, and it was decreased by the administration of either ECE or DK.

The urine volume of the SHRs progressively decreased as compared with that of the WKY rats of the same age [38]. It is known that the greater the excretion of urine sodium, the less the urinary potassium excretion. Moreover, in human studies, the urine Na/K ratio is associated with the increase in BP and the fast impairment of renal function [39]. In our study, the urine volume of the SHRs was lower than that of the WKY rats, even though the water intake amount was not different between the SHRs and WKY rats. The urine Na/K ratio of the SHRs was significantly different among the WKY rats, SHRs, and ECE- or DK-treated groups. The urine Na/K ratio was used as a marker of dietary sodium and potassium intake in human studies [39]. The BP increased with the increase in sodium intake. However, increased potassium intake decreased the BP by increasing the excretion of sodium into the urine [40]. Therefore, the urine Na/K ratio is associated with the BP in humans [40]. In our study, all animals were not fed with a high-sodium diet, which explains why the urine Na/K ratio among the groups did not show any significant difference.

Hypertension tends to increase proteinuria [37]. It is known that the urine albumin-to-creatinine ratio is associated with the decrease in the glomerular filtration rate [41]. Previous studies have shown that the development of microalbuminuria in SHRs is caused by predominant tubular injury, which induce the urinary loss of low molecular weight proteins [42,43]. In our study, the urine albumin-to-creatinine ratio of the SHRs was higher than that of the WKY rats, and was decreased by the administration of either ECE or DK.

Previous studies showed that several polyphenols have an inhibiting effect on ACE activity [14,15,16,17]. However, these studies only reported ACE inhibitory activity as an IC50 value, which is the 50% inhibition concentration of a positive control, and they did not show an antihypertensive effect or protecting effect against hypertensive nephropathy in the animals. One study showed that ECE decreased BP in the 2-kindney 1-clip Goldblatt hypertensive rats [44]. Pyrogallol-phloroglucinol-6,6-bieckol decreased BP in the high fat diet-induced hypertension animal model by modulating vascular dysfunction [45]. However, these studies showed only decreasing BP and did not evaluate whether ECE showed any effect of attenuating hypertensive nephropathy. Our study showed that ECE and DK decreased the expression of AT1R/TGFβ/SMAD2/3 in the kidney and decreased the serum level of Ang II. Additionally, ECE and DK decreased EMT, renal fibrosis and glomerular sclerosis. It seems that ECE and DK might be helpful in attenuating hypertensive nephropathy.

## 4. Materials and Methods

### 4.1. Preparation of ECE and Isolation of DK

The ECE were obtained from the Aqua Green Technology Co., Ltd. (Jeju, Korea). The E. cava was thoroughly washed with pure water and air-dried at room temperature for 48 h. It was ground, and 50% ethanol was added, followed by heating at 85 °C for 12 h. The ECE were filtered and then, when concentrated, sterilized by heating at a high temperature for 40–60 min and then spray-dried. Dieckol, which is one of the representative phlorotannins of E. cava, was then isolated. Briefly, centrifugal partition chromatography was performed using a two-phase solvent system that mixed pure water, ethyl acetate, n-hexane, and a methanol mixture (ratio 7:7:2:3). The organic stationary phase was filled in the column and the mobile phase was filled in the column, following the descending order of flow rate (2 mL per minute) and was used for separation. We followed the methods of Dr. Son et al. (2019) [45].

### 4.2. Hypertension Animal Model

Male SHRs (8 weeks old) and male WKY rats (8 weeks old) were bought from the Orient Bio (Sungnam, Korea) and kept at a constant temperature of approximately 24 °C, relative humidity of 50–55%, and light/dark cycle of 12 h/12 h. The rat sublimation was conducted for 1 week. The rats were randomly divided into six groups (five rats per group):

(i) WKY group, in which the rats were orally administered with drinking water for 4 weeks;

(ii) SHR group, in which the rats were orally administered with drinking water for 4 weeks;

(iii–v) SHR group, in which the rats were orally administered with ECE of (iii) 50 mg/kg/day, (iv) 100 mg/kg/day, and (v) 150 mg/kg/day for 4 weeks;

(vi) SHR group, in which the rats were orally administered with 2.5 mg/kg/day DK for 4 weeks. After 4 weeks, water consumption, urine volume and urine chemical analysis were validated with metabolic cage for 24 h. After completing the metabolic analysis, all rats were sacrificed according to the ethical principles of the Institutional Animal Care and Use Committee of Gachon University (approval number: LCDI-2019-0121).

### 4.3. Immunohistochemistry

Kidney tissue paraffin blocks were sectioned at 8 µm, placed on gelatin-coating slides, and dried at 37 °C for 3 days. The kidney tissue slides were deparaffinized and then incubated with 0.3% hydrogen peroxide (Sigma-Aldrich, MO, USA) for 10 min. Afterward, the slides were rinsed three times with phosphate-buffered saline (PBS), and incubated with an animal serum to inhibit antibody nonspecific binding. Moreover, they were incubated with primary antibodies (anti-AT1R, anti-E-cadherin, anti-α-SMA, and anti-vimentin) and rinsed with PBS three times. The tissue slides were then incubated with the biotinylated secondary antibodies from the ABC kit (Vector Laboratories, Burlingame, CA, USA), incubated for 3 h with a blocking solution, and rinsed with PBS three times. The tissue slides were developed with 3,3-diaminobenzidine as a substrate for 3 min, mounted with a xylene-based DPX solution (Sigma-Aldrich), and visualized via light microscopy (Olympus Optical Co., Tokyo, Japan).

### 4.4. RNA Extraction and Quantitative Real-Time Polymerase Chain Reaction (qRT-PCR)

The RNA from the rat kidney tissues was isolated using the RNAiso Plus reagent (TAKARA, Japan) according to the manufacturer’s instructions. The kidney tissues were first divided into two parts, namely, the cortex part and the medulla part. The kidney tissues were finely cut and crushed into powder, and the powders were resuspended with 1 mL of RNAiso Plus reagent, mixed with 0.1 mL of pure glade chloroform (Amresco, Solon, OH, USA), and then centrifuged at 13,000× *g* for 20 min at 4 °C. After centrifugation, the supernatant was mixed with 0.25 mL of pure glade isopropyl alcohol, and the extracted RNA pellets were washed with 70% alcohol and centrifuged at 7500× *g* for 5 min at 4 °C. The dried pellets were dissolved in 10–30 µl diethylpyrocarbonate-treated pure water, and RNA was quantified and checked using the NanoDrop 2000 (Thermo Fisher Scientific, MA, USA). Complementary DNA (cDNA) was prepared from RNA using a cDNA synthesis kit (PrimeScript™, TAKARA). A quantitative real-time polymerase chain reaction (qRT-PCR) determined the RNA levels from the kidney tissues. The forward and reverse primers were mixed with distilled water, and then 10 µl mixtures were placed in a 384-well qRT-PCR plate. The cDNA and SYBR green (TAKARA) were subsequently added and then validated using a qRT-PCR machine (Bio-Rad, Hercules, CA, USA). The genes of interest are listed in Appendix A. We followed the methods of Dr. Son et al. 2019 [45].

### 4.5. Enzyme-Linked Immunosorbent Assay (ELISA)

To confirm the serum Ang II level, an aliquot of the withdrawn blood (1.5 to 1.7 mL) was added in serum separator tubes (Becton Dickinson, Franklin Lakes, NJ, USA) and then centrifuged at 2000× *g* for 20 min at room temperature. Afterward, the separated serum was moved into a new tube and stored in a deep freezer. The Ang II ELISA kit (MyBioSource, San Diego, CA, USA) was used for analyzing the presence of serum according to the manufacturer’s instructions. Ang II was measured within 1 week after blood collection from the animals.

### 4.6. Histological Analysis

#### 4.6.1. Masson’s Trichrome (MT) Stain

Masson’s trichrome (MT) stain validates the fibrosis of the kidney. Blocks of paraffin-embedded kidney tissue were sectioned to a thickness of 8 µm, placed on a coating slide, and dried at 37 °C for 3 days. The tissue slides were deparaffinized with xylene and alcohol and were re-fixed with Bouin’s solution for 24 h and rinsed with running tap water for 3 min. The slides were submerged with Weigert’s iron hematoxylin solution for 10 min and Biebrich scarlet-acid fuchsin solution for 15 min and then differentiated with phosphomolybdic–phosphotungstic acid solution for 10 min. Finally, the slides were transferred to aniline blue solution for 3 min and then washed with water. The stained slides were mounted with xylene-based DPX solution (Sigma-Aldrich) and visualized via light microscopy (Olympus). The collagen fiber appears to be blue in color in the fibrosis area, whereas the renal epithelium appears to be red in color. MT-stained fibrosis areas were measured using the ImageJ software. Original images of the MT-stained kidney were converted into RGB images, and then these images were deconvolved using the ImageJ software using the color deconvolution plugin.

#### 4.6.2. Periodic Acid–Schiff (PAS) Stain

A periodic acid–Schiff (PAS) stain validated the glomerular damage of the kidney. Blocks of paraffin-embedded kidney tissue were sectioned to a thickness of 8 µm, placed on a coating slide, and dried at 37 °C for 3 days. The tissue slides were deparaffinized with xylene and alcohol, hydrated with water, oxidized with 0.5% periodic acid solution for 5 min, and washed with water. The slides were submerged with the Schiff reagent for 15 min and then washed with lukewarm tap water for 5 min. Finally, the slides were transferred to Mayer’s hematoxylin solution for 1 min and then washed with water. The stained slides were mounted with xylene-based DPX solution (Sigma-Aldrich) and visualized via light microscopy (Olympus).

Glomeruli were randomly selected, and the glomerular damage was evaluated using a semi-quantitative scoring method (Grades 0–4) using PAS-stained kidney tissue slides.

(1)Grade 0: normal glomeruli;(2)Grade 1: minimal sclerosis (sclerotic area up to 25%);(3)Grade 2: moderate sclerosis (sclerotic area 25% to 50%);(4)Grade 3: moderate–severe sclerosis (sclerotic area 50% to 75%);(5)Grade 4: severe sclerosis (sclerotic area 75% to 100%);

The glomerulosclerotic index (GSI) was calculated using the following equation: (4 × n4) + (3 × n3) + (2 × n2) + (1 × n1) / n4 + n3 + n2 + n1 + n0, where n (x) is the number of renal glomerular sclerosis in each grade [46]. This analysis was carried out in the treatment groups by a masked observer.

### 4.7. Systolic Blood Pressure, Diastolic Blood Pressure, and Mean Arterial Blood Pressure Measurements

The systolic BP, diastolic BP, and mean arterial BP were measured using a noninvasive CODA tail-cuff system (Kent Scientific Corp., Torrington, CT, USA). The rat sublimation was conducted for 20 min for 5 days, and the BP was measured on the last day.

### 4.8. In Vitro Modeling Using TCMK-1 Cells

TCMK-1 cells, a mouse proximal tubule cell line, were purchased from American Type Culture Collection (Washington, DC, USA). High glucose Dulbecco’s Modified Eagle’s medium (Gibco; Grand island, NY, USA), 10% fetal bovine serum (FBS) and 1% penicillin-streptomycin were used as culture mediums. To validate the inhibitory effects of ECE and DK in angiotensin II-treated TCMK-1 cells, we treated ECE (5, 25, 50 μg/mL) and DK (1.8 μg/mL) with 100 nmol/L angiotensin II (Sigma-Aldrich, St Louis, MO, USA) for 12 h. Telmisartan (1 μmol/L, Sigma-Aldrich) was used for AT1 receptor antagonist and pretreated before angiotensin II for 3 h [47].

### 4.9. Preparation of Protein and Western Blotting

To isolate protein from angiotensin II-treated TCMK-1 cells, the cells were scraped using the RIPA lysis buffer with phosphatase and proteinase inhibitor (EzRIPA; ATTO; Tokyo, Japan) and incubated on ice for 20 min. After centrifuging at 13,000× *g* for 20 min, at 4 °C, clean supernatants were moved to a new tube and the concentration of supernatants was analyzed with a bicinchoninic acid assay kit (BCA kit; Thermo Fisher Scientific, Inc.; Waltham, MA, USA). To validate protein expression from TCMK-1 cells, blotting was conducted. An equal amount of lysate proteins (30 μg/lane) was separated by 10% sodium dodecyl sulfate polyacrylamide gel using electrophoresis. Then, running proteins were transferred to polyvinylidene fluoride membranes, which were incubated with diluted primary antibodies (Anti-β-actin, AGTR1, TGF-β, SMAD2/3 and pSMAD2/3) at 4 °C. The incubated membranes with antibodies were thoroughly washed using Tris-buffered saline with 0.1% Tween 20 and incubated with secondary antibodies for 2 h at room temperature. All membranes were developed by enhanced chemiluminescence (LAS-4000s; GE Healthcare, Chicago, IL, USA). The antibodies information used in this study can be found in Appendix A.

### 4.10. Statistical Analysis

The results are presented as mean ± SD, and *p*-values of <0.05 means it is statistically significant. The Kruskal–Wallis test was used to validate the differences among the groups, and the Mann–Whitney U test was used in the SPSS ver. 22 software (IBM Corporation; NY, Armonk, USA) completed post hoc comparisons. Means denoted by a different letter indicate significant differences between groups.

## 5. Conclusions

In conclusion, AT1R was upregulated in the SHRs, and the signal pathway increased TGFβ and SMAD2 or 3, which induced EMT. Either ECE or DK downregulated the signal pathway and decreased EMT. Therefore, ECE or DK decreased the mesenchymal cell markers such as the expression of vimentin and α-SMA, the fibrotic tissue of the kidney, and the ratio of urine albumin/creatinine and attenuated the EMT. Furthermore, renal fibrosis led to the restoration of renal function (Figure 7).

## Figures and Tables

**Figure 1 ijms-22-04230-f001:**
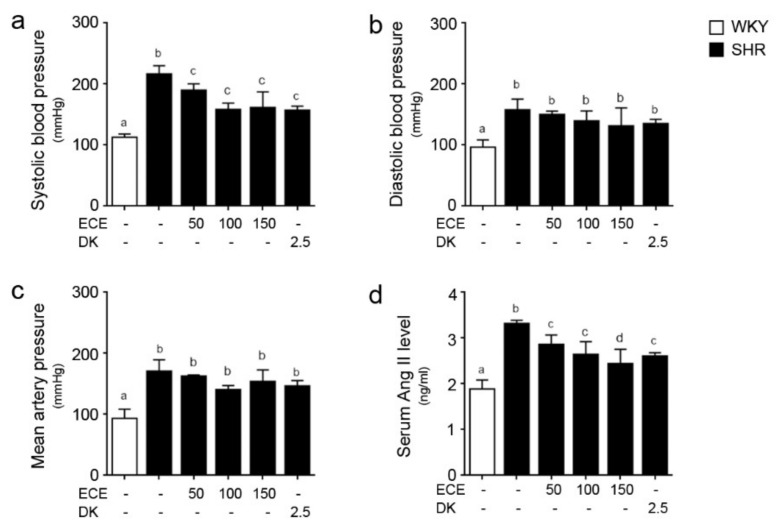
Comparative analysis of ECE and DK administration on the reduction of systolic blood pressure and serum angiotensin II level in SHRs. (**a**) Systolic blood pressure, (**b**) diastolic blood pressure, (**c**) mean artery blood pressure, and (**d**) serum angiotensin II level were measured prior to sacrifice. Three doses of ECE (50 mg/kg/day, 100 mg/kg/day and 150 mg/kg/day) were oral administrated for 4 weeks and 2.5 mg/kg/day DK also oral administrated for 4 weeks. Means denoted by a different letter indicate significant differences among groups (*p* < 0.05). - means the same group. ECE, Ecklonia cava extract; DK, dieckol.

**Figure 2 ijms-22-04230-f002:**
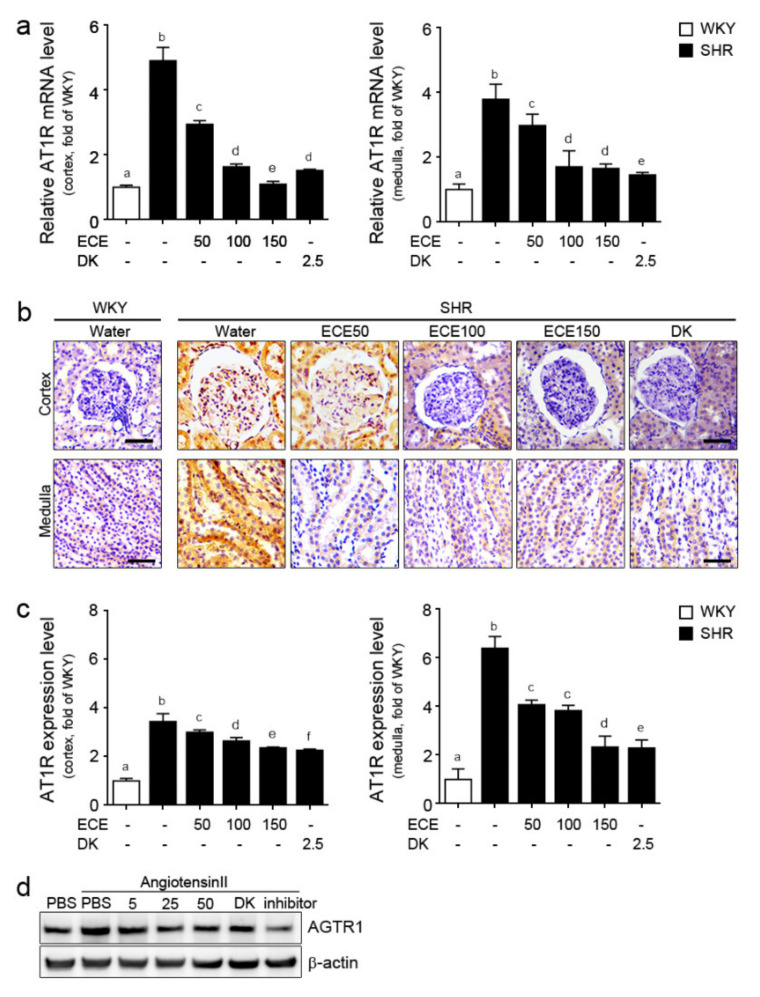
Comparative analysis of ECE and DK administration on the reduction of expression of AT1R in the kidney of SHRs and in angiotensin II-treated TCMK-1 cells. (**a**) AT1R mRNA levels in cortex area, and in medulla of the kidney were measured by qRT-PCR analysis. (**b**) AT1R protein expression levels in cortex area, and in medulla of kidney were measured by immunohistochemistry and (**c**) the protein levels were quantified by Image J software. Three doses of ECE (50 mg/kg/day, 100 mg/kg/day and 150 mg/kg/day) were orally administrated for 4 weeks and 2.5 mg/kg/day of DK were also orally administrated for 4 weeks. (**d**) AT1R protein expression levels in angiotensin II-treated TCMK-1 cells (renal epithelial tubular cells) with 5, 25, 50 ug/mL ECE, 2.5 ug/mL DK or inhibitor were measured by Western blotting. Scale bar = 25 μm. Means denoted by a different letter indicate significant differences among groups (*p* < 0.05). - means the same group. ECE, Ecklonia cava extract; DK, dieckol; Inhibitor, 1 μmol/L telmisartan.

**Figure 3 ijms-22-04230-f003:**
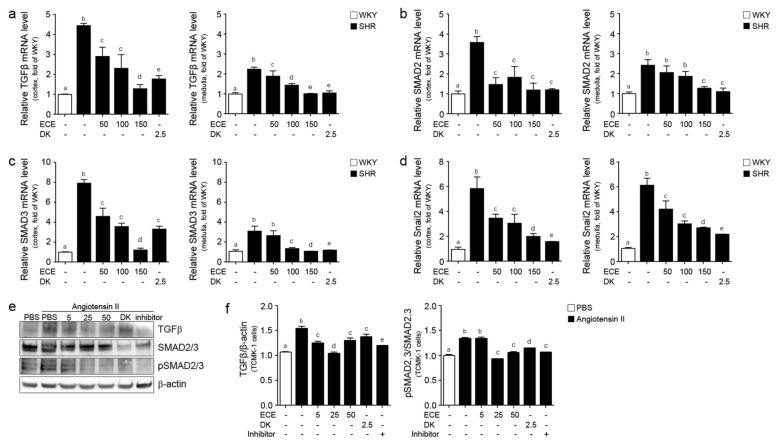
Comparative analysis of ECE and DK administration on the reduction of expression of TGFβ, SMAD2/3, and Snail2 in the kidneys of SHRs. (**a**) TGFβ, (**b**) SMAD2, (**c**) SMAD3, (**d**) Snail2 mRNA levels in cortex area, and in medulla of kidney were measured by qRT-PCR analysis. Three doses of ECE (50 mg/kg/day, 100 mg/kg/day and 150 mg/kg/day) were orally administrated for 4 weeks and 2.5 mg/kg/day DK were also orally administrated for 4 weeks. (**e**) TGFβ, SMAD2/3, pSMAD2/3 protein expression levels in angiotensin II-treated TCMK-1 cells (renal tubular epithelial cells) with 5, 25, 50 ug/mL ECE, 2.5 ug/mL DK or inhibitor were measured by Western blotting and (**f**) the protein levels were quantified by Image J software. Means denoted by a different letter indicate significant differences between groups (*p* < 0.05). - means the same group. ECE, Ecklonia cava extract; DK, dieckol; Inhibitor, 1 μmol/L telmisartan.

**Figure 4 ijms-22-04230-f004:**
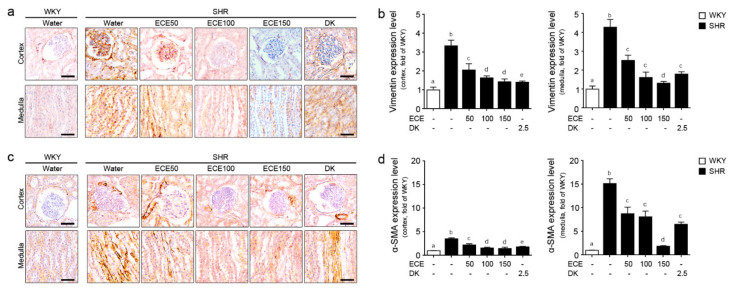
Comparative analysis of ECE and DK administration on the reduction of EMT in the kidney of the SHRs. (**a**) Vimentin protein expression levels in cortex area, and in medulla of kidney were measured by immunohistochemistry and (**b**) the protein levels were quantified by Image J software. (**c**) α-SMA protein expression levels in cortex area, and in medulla of kidney were measured by immunohistochemistry and (**d**) the protein levels were quantified by Image J software. Scale bar = 25 μm. Three doses of ECE (50 mg/kg/day, 100 mg/kg/day and 150 mg/kg/day) was oral administrated for 4 weeks and 2.5 mg/kg/day DK also oral administrated for 4 weeks. Means denoted by a different letter indicate significant differences between groups (*p* < 0.05). - means the same group. ECE, Ecklonia cava extract; DK, dieckol.

**Figure 5 ijms-22-04230-f005:**
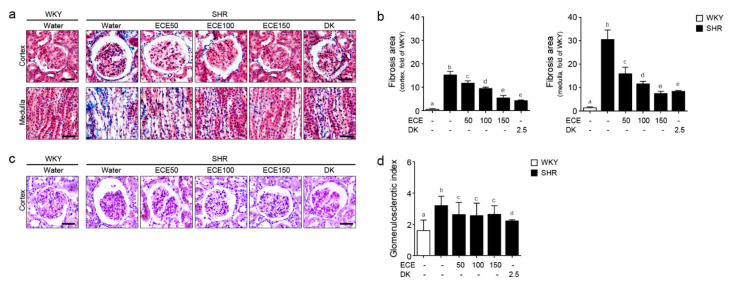
Comparative analysis of ECE and DK administration on the reduction of renal fibrosis and glomerular sclerosis in the kidney of SHRs. (**a**) Masson’s trichrome stained cortex area and medulla area of kidney showing fibrosis area (blue color) and (**b**) fibrosis area was quantified by Image J software. (**c**) PAS-stained cortex area and medulla area of kidney showing glomerular sclerosis and (**d**) the sclerosis were evaluated using a semi-quantitative scoring method (Grades 0–4). Scale bar = 25 μm. Three doses of ECE (50 mg/kg/day, 100 mg/kg/day and 150 mg/kg/day) were orally administrated for 4 weeks and 2.5 mg/kg/day of DK was also orally administrated for 4 weeks. Means denoted by a different letter indicate significant differences between groups (*p* < 0.05). - means the same group. ECE, Ecklonia cava extract; DK, dieckol; GSI; glomerulosclerotic index.

**Figure 6 ijms-22-04230-f006:**
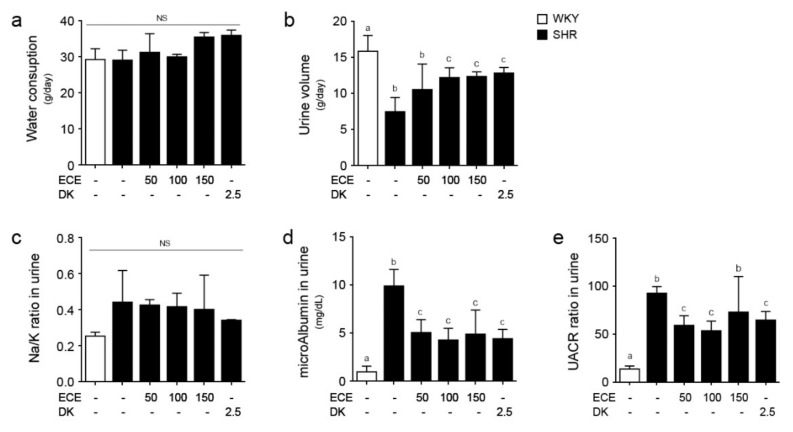
Comparative analysis of ECE and DK administration on the attenuation of renal function aggravation in SHRs. (**a**) Water consumption, (**b**) urine volume, (**c**) urine Na/K ratio, (**d**) urine microalbumin level, (**e**) urine albumin-to-creatinine ratio (UACR) were measured prior to sacrifice. Three doses of ECE (50 mg/kg/day, 100 mg/kg/day and 150 mg/kg/day) were orally administrated for 4 weeks and 2.5 mg/kg/day of DK was also orally administrated for 4 weeks. Means denoted by a different letter indicate significant differences between groups (*p* < 0.05). - means the same group. ECE, Ecklonia cava extract; DK, dieckol; NS, not significant.

**Figure 7 ijms-22-04230-f007:**
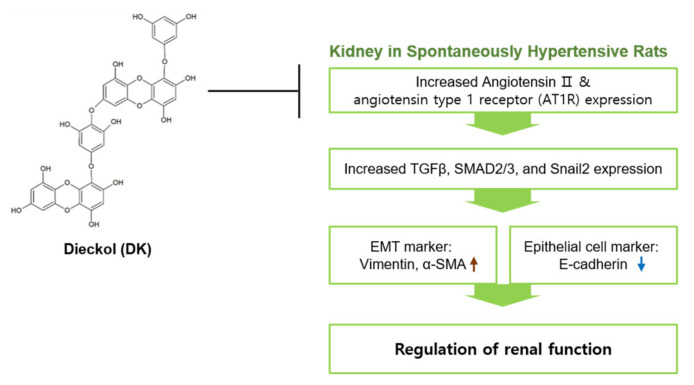
Schematic image for effects of ECE and DK administration in SHRs. ECE or DK downregulated AT1R/TGFβ/SMAD2 or three molecule expression and EMT marker expression in SHRs.

## Data Availability

All data are contained within the article.

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
