# Peer review of "Attenuating Effects of Dieckol on Hypertensive Nephropathy in Spontaneously Hypertensive Rats"

_ijms, 2021, doi:10.3390/ijms22084230_

Round 1

Reviewer 1 Report

no further comments

Author Response

We respect your opinion. Thank you.

Reviewer 2 Report

With this revised manuscript, the authors have responded adequately to my initial concerns. However, I suggest some minor concerns. 
1. Urine creatinine level is not a representative marker of renal function. Please remove the result of urine creatinine levels in figure 6F. I recommend to evaluate serum creatinine level in the future experiments.
 2. The protein expressions associated with signaling pathway were not included in the revised manuscript.  The confirmation of protein expressions are very important in the experiments which elucidate the molecular pathway. I recommend to include the results of western blots in the future experiments.

Author Response

Response to Reviewer 2 Comments

With this revised manuscript, the authors have responded adequately to my initial concerns. However, I suggest some minor concerns. 
Point 1: Urine creatinine level is not a representative marker of renal function. Please remove the result of urine creatinine levels in figure 6F. I recommend to evaluate serum creatinine level in the future experiments.

Response 1: We appreciated with this comment. As your comment, urine creatinine result removed in figure 6F and we will evaluate serum creatinine level in the future study.

Figure and its legend (previous version)

Figure 6. Comparative analysis of ECE and DK administration on the attenuation of renal function aggravation in SHRs. (a) water consumption, (b) urine volume, (c) urine Na/K ratio, (d) urine microalbumin level, (e) urine albumin-to-creatinine ratio (UACR) and (f) urine creatine levels were measured prior to sacrifice. Three doses of ECE (50 mg/kg/day, 100 mg/kg/day and 150 mg/kg/day) were oral administrated for 4 weeks and 2.5 mg/kg/day DK also oral administrated for 4 weeks. Means denoted by a different letter indicate significant differences between groups (p < 0.05). ECE, Ecklonia cava extract; DK, dieckol.

Figure and its legend (new version)

Figure 6. Comparative analysis of ECE and DK administration on the attenuation of renal function aggravation in SHRs. (a) water consumption, (b) urine volume, (c) urine Na/K ratio, (d) urine microalbumin level, (e) urine albumin-to-creatinine ratio (UACR) were measured prior to sacrifice. Three doses of ECE (50 mg/kg/day, 100 mg/kg/day and 150 mg/kg/day) were oral administrated for 4 weeks and 2.5 mg/kg/day DK also oral administrated for 4 weeks. Means denoted by a different letter indicate significant differences between groups (p < 0.05). ECE, Ecklonia cava extract; DK, dieckol.

Results section (previous version)

2.6. ECE and DK attenuated renal function aggravation in SHRs

The amount of water intake within 24 hours was not significantly different among all groups (Figure 6a). The urine volume of the SHRs within 24 hours was significantly lower than that of the WKY rats (Figure 6b). It was significantly increased by the administration of 100 and 150 mg/kg/day of ECE and DK. The urine sodium/potassium (Na/K) ratio was not significantly different among all groups (Figure 6c). The albumin level in the urine of the SHRs was significantly higher than that of the WKY rats and was significantly decreased by the administration of ECE or DK. The decreasing effects of 50, 100, and 150 mg/kg/day of ECE and DK were not significantly different (Figure 6d). The ratio of urine albumin/creatinine and urine creatine levels of the SHRs was significantly higher than that of the WKY rats and was decreased by the administration of ECE or DK. The decreasing effects of 50 and 100 mg/kg/day of ECE and DK were not significantly different (Figure 6e and f).

Results section (new version)

2.6. ECE and DK attenuated renal function aggravation in SHRs

The amount of water intake within 24 hours was not significantly different among all groups (Figure 6a). The urine volume of the SHRs within 24 hours was significantly lower than that of the WKY rats (Figure 6b). It was significantly increased by the administration of 100 and 150 mg/kg/day of ECE and DK. The urine sodium/potassium (Na/K) ratio was not significantly different among all groups (Figure 6c). The albumin level in the urine of the SHRs was significantly higher than that of the WKY rats and was significantly decreased by the administration of ECE or DK. The decreasing effects of 50, 100, and 150 mg/kg/day of ECE and DK were not significantly different (Figure 6d). The ratio of urine albumin/creatinine of the SHRs was significantly higher than that of the WKY rats and was decreased by the administration of ECE or DK. The decreasing effects of 50 and 100 mg/kg/day of ECE and DK were not significantly different (Figure 6e).

Point 2: The protein expressions associated with signaling pathway were not included in the revised manuscript. The confirmation of protein expressions are very important in the experiments which elucidate the molecular pathway. I recommend to include the results of western blots in the future experiments.

Response 2: We appreciated with this comment. As your comment, we will include signaling pathway related western blot results in the future study.

Reviewer 3 Report

The authors showed considerable improvement and given rationales to the article, however, the authors should consider the followings:

  1. In section 2.2, the authors should give rationale(s) of why TCMK-1 cells were selected for the in vitro model, of the current study; the authors should provide relevant references to support that.
  2. In section 4.9, please give rationales for using beta-actin as house-keeping in your research/disease-area. Sometimes, the actin network maybe triggered in specific disease model.
  3. In section 4.8, please explain and provide evidences of using the dose and duration of Ang II.
  4. In Figure 3e, the authors may add extra graph to show quantitatively by densitometry analysis.
  5. In section 4.5, the authors should state the stability limit of serum, and how long did the authors store the serum before testing.
  6. In section 4.9, the authors should state the clone number, dilution (or conc. of) the primary antibodies used.
  7. In the conflict of interest, please state whether the interpretation of results in the study has any influences by the sponsoring/ commercial partners.

Author Response

Response to Reviewer 3 Comments

The authors showed considerable improvement and given rationales to the article, however, the authors should consider the followings:

Point 1: In section 2.2, the authors should give rationale(s) of why TCMK-1 cells were selected for the in vitro model, of the current study; the authors should provide relevant references to support that.

Response 1: We appreciated with this comment. TCMK-1 cells are mouse renal tubular epithelial cells and it is commonly used in the study of renal injury [Reference; Cell Physiol Biochem. 2018;45(6):2268-2282.; Front Med (Lausanne). 2020 Jan 24;7:5.; Mar Drugs. 2019 Oct 24;17(11):602.]. As this comment, the reference was added in Results and Reference section.

Results section (previous version)

2.2. ECE and DK attenuated the expression of AT1R in the kidney of SHRs

The expression of AT1R was evaluated through qRT-PCR and staining. The mRNA expression of AT1R in the medulla and cortex of the kidney was significantly higher in the SHRs than that of the WKY rats. Such expression was decreased by the administration of either ECE or DK (Figure 2a). In the medulla, the decreasing effect was most prominent in the group treated with DK. In the cortex, 150 mg/kg/day of ECE had the most prominent decreasing effect. The expression level of AT1R in the cortex and medulla of the SHRs, which was evaluated through staining, was significantly increased and was significantly decreased by the administration of either ECE or DK. Moreover, the most prominent decreasing effect was observed in the group treated with DK (Figure 2b and c). In in vitro model, the expression level of AT1R protein in angiotensin â…¡-treated TCMK-1 cells with ECE (5, 25, 50 ug/ml), DK or inhibitor was decreased than only angiotensin â…¡ treatment (Figure 2d).

Results section (new version)

2.2. ECE and DK attenuated the expression of AT1R in the kidney of SHRs

The expression of AT1R was evaluated through qRT-PCR and staining. The mRNA expression of AT1R in the medulla and cortex of the kidney was significantly higher in the SHRs than that of the WKY rats. Such expression was decreased by the administration of either ECE or DK (Figure 2a). In the medulla, the decreasing effect was most prominent in the group treated with DK. In the cortex, 150 mg/kg/day of ECE had the most prominent decreasing effect. The expression level of AT1R in the cortex and medulla of the SHRs, which was evaluated through staining, was significantly increased and was significantly decreased by the administration of either ECE or DK. Moreover, the most prominent decreasing effect was observed in the group treated with DK (Figure 2b and c).

To validate ECE and DK attenuation of AT1R expression in kidney tubule, an mouse proximal tubule cell line (TCMK-1) was activated by angiotensin â…¡ in in vitro model [20]. The expression level of AT1R protein in angiotensin â…¡-treated TCMK-1 cells with ECE (5, 25, 50 ug/ml), DK or inhibitor was decreased than only angiotensin â…¡ treatment (Figure 2d).

Reference section (new version)

Son, M.; Oh, S.; Choi, C. H.; Park, K. Y.; Son, K. H.; Byun, K. Pyrogallol-Phloroglucinol-6,6-Bieckol from Ecklonia cava Attenuates Tubular Epithelial Cell (TCMK-1) Death in Hypoxia/Reoxygenation Injury. Mar drugs. 2019, 17, 602, https://doi.org/10.3390/md17110602

Point 2: In section 4.9, please give rationales for using beta-actin as house-keeping in your research/disease-area. Sometimes, the actin network maybe triggered in specific disease model.

Response 2: We appreciated with this comment. We agreed with this comment. As this comment, beta-actin, actin or GAPDH can be used as house-keeping gene. We used beta-actin is as house-keeping in SHR animal. Some reference papers also used beta-actin as house-keeping in SHR animal [References: Free Radic Res. 2009 Dec;43(12):1195-204.; Cardiovasc Res. 2016 Nov 1;112(2):581-589.].

Point 3: In section 4.8, please explain and provide evidences of using the dose and duration of Ang II.

Response 3: We appreciated with this comment. To make in vitro model using TCMK-1 cells, we refer to the paper below [Reference: Acta Pharmacol Sin. 2015 Jul;36(7):821-30.]. Briefly, 100 nmol/L angiotensin â…¡-treated TCMK-1 cells with or without (5, 25, 50 μg/mL) and DK (1.8 μg/mL) for 12 hours. After 12 hours, protein extract from the cells. To make inhibitor model, 1 μmol/L Telmisartan, which is AT1 receptor antagonist, pre-teated before angiotensin â…¡ for 3 hours. Reference paper of using the dose and duration of Ang II added in material and methods section and reference section.

Material and methods section (previous version)

4.8. In vitro modeling using TCMK-1 cells

TCMK-1 cells, a mouse proximal tubule cell line, were purchased from American Type Culture Collection (Washington, DC, USA). High glucose Dulbecco's Modified Eagle's medium (Gibco; Grand island, NY, USA), 10% fetal bovine serum (FBS) and 1% penicillin-streptomycin were used as culture medium. To validate the inhibitory effects of ECE and DK in angiotensin â…¡-treated TCMK-1 cells, we treated ECE (5, 25, 50 μg/mL) and DK (1.8 μg/mL) with 100 nmol/L angiotensin â…¡ (Sigma-Aldrich, St Louis, MO, USA) for 12 hours. Telmisartan (1 μmol/L, Sigma-Aldrich) was used for AT1 re-ceptor antagonist and pretreated before angiotensin â…¡ for 3 hours.

Material and methods section (new version)

4.8. In vitro modeling using TCMK-1 cells

TCMK-1 cells, a mouse proximal tubule cell line, were purchased from American Type Culture Collection (Washington, DC, USA). High glucose Dulbecco's Modified Eagle's medium (Gibco; Grand island, NY, USA), 10% fetal bovine serum (FBS) and 1% penicillin-streptomycin were used as culture medium. To validate the inhibitory effects of ECE and DK in angiotensin â…¡-treated TCMK-1 cells, we treated ECE (5, 25, 50 μg/mL) and DK (1.8 μg/mL) with 100 nmol/L angiotensin â…¡ (Sigma-Aldrich, St Louis, MO, USA) for 12 hours. Telmisartan (1 μmol/L, Sigma-Aldrich) was used for AT1 re-ceptor antagonist and pretreated before angiotensin â…¡ for 3 hours [49].

Reference (new version)

Wang, J.; Wen, Y.; Lv, L.L.; Liu, H.; Tang, R.N.; Ma, K.L.; Liu, B.C. Involvement of endoplasmic reticulum stress in angiotensin II-induced NLRP3 inflammasome activation in human renal proximal tubular cells in vitro. Acta Pharmacol Sin. 2015, 36, 821-830.

Point 4: In Figure 3e, the authors may add extra graph to show quantitatively by densitometry analysis.

Response 4: We appreciated with this comment. As this comment, quantitative graphs from western blot results (Figure 3e) added in Figure 3. Figure 3, its legend and Results section of manuscript was edited.

Figure and its legend (previous version)

Figure 3. Comparative analysis of ECE and DK administration on the reduction of expression of TGFβ, SMAD2/3, and Snail2 in the kidney of SHRs. (a) TGFβ, (b) SMAD2, (c) SMAD3, (d) Snail2 mRNA levels in cortex area, and in medulla of kidney were measured by qRT-PCR analysis. Three doses of ECE (50 mg/kg/day, 100 mg/kg/day and 150 mg/kg/day) was oral administrated for 4 weeks and 2.5 mg/kg/day DK also oral administrated for 4 weeks. (e) TGFβ, SMAD2/3, pSMAD2/3 protein expression levels in angiotensin â…¡-treated TCMK-1 cells with 5, 25, 50 ug/ml ECE, 2.5 ug/ml DK or inhibitor were measured by Western blotting. Means denoted by a different letter indicate significant differences between groups (p < 0.05). ECE, Ecklonia cava extract; DK, dieckol, Inhibitor, 1 μmol/L telmisartan.

Figure and its legend (new version)

Figure 3. Comparative analysis of ECE and DK administration on the reduction of expression of TGFβ, SMAD2/3, and Snail2 in the kidney of SHRs. (a) TGFβ, (b) SMAD2, (c) SMAD3, (d) Snail2 mRNA levels in cortex area, and in medulla of kidney were measured by qRT-PCR analysis. Three doses of ECE (50 mg/kg/day, 100 mg/kg/day and 150 mg/kg/day) was oral administrated for 4 weeks and 2.5 mg/kg/day DK also oral administrated for 4 weeks. (e) TGFβ, SMAD2/3, pSMAD2/3 protein expression levels in angiotensin â…¡-treated TCMK-1 cells with 5, 25, 50 ug/ml ECE, 2.5 ug/ml DK or inhibitor were measured by Western blotting and (f) the protein levels were quantified by Image J software. Means denoted by a different letter indicate significant differences between groups (p < 0.05). ECE, Ecklonia cava extract; DK, dieckol, Inhibitor, 1 μmol/L telmisartan.

Results (new version)

2.3. ECE and DK reduced the expression of TGFβ, SMAD2/3, and Snail2 in the kidney of SHRs

The expressions of TGFβ in the medulla and cortex of the SHRs were significantly higher than those of the WKY rats, which were decreased by the administration of ECE or DK. In the medulla, the decreasing effect was most prominent in the groups treated with 150 mg/kg/day of ECE and DK. In the cortex, the decreasing effect was most prominent in the group treated with DK (Figure 3a). The expressions of SMAD2 in the medulla and cortex of the SHRs were higher than those of the WKY rats, which were significantly decreased by the administration of ECE or DK. The decreasing effect in the medulla was most prominent in the groups treated with 150 mg/kg/day of ECE or DK. In the cortex, the decreasing effects of 50, 100, and 150 mg/kg/day of ECE or DK were not significantly different (Figure 3b). The expression of SMAD3 in the medulla and cortex of the SHRs was significantly higher than that of the WKY rats and was significantly decreased by the administration of ECE or DK. The most prominent decreasing effect was observed in the group treated with 150 mg/kg/day of ECE (Figure 3c). The expression of Snail2 in the medulla and cortex of the SHRs was significantly increased by the administration of ECE or DK. The most prominent decreasing effect was observed in the group treated with DK (Figure 3d). In in vitro model, the expression level of TGFβ and pSMAD2/3 protein in angiotensin â…¡-treated TCMK-1 cells with ECE (5, 25, 50 ug/ml), DK or inhibitor was decreased than only angiotensin â…¡ treatment (Figure 3e and f).

Point 5: In section 4.5, the authors should state the stability limit of serum, and how long did the authors store the serum before testing.

Response 5: We appreciated with this comment. As your opinion, in order to maintain stability in serum, we isolated serum from whole blood, immediately immersed in liquid nitrogen (LN2), and stored in deep freezer. In addition, Angiotensin â…¡ level was measured within 1 week after collection. This information added in material and methods section.

Material and methods section (previous version)

4.5. Enzyme-linked immunosorbent assay (ELISA)

To confirm the serum Ang II level, an aliquot of the withdrawn blood (1.5 to 1.7 ml) was added in serum separator tubes (Becton Dickinson, NJ, USA) and then centrifuged at 2,000 × g for 20 minutes at room temperature. Afterward, the separated serum was moved into a new tube and stored in a deep freezer. The Ang II ELISA kit (MyBioSource, CA, USA) was used for analyzing the presence of serum according to the manufacturer’s instructions.

Material and methods section (new version)

4.5. Enzyme-linked immunosorbent assay (ELISA)

To confirm the serum Ang II level, an aliquot of the withdrawn blood (1.5 to 1.7 ml) was added in serum separator tubes (Becton Dickinson, NJ, USA) and then centrifuged at 2,000 × g for 20 minutes at room temperature. Afterward, the separated serum was moved into a new tube and stored in a deep freezer. The Ang II ELISA kit (MyBioSource, CA, USA) was used for analyzing the presence of serum according to the manufacturer’s instructions. Ang II was measured within 1 week after blood collection from animals.

Point 6: In section 4.9, the authors should state the clone number, dilution (or conc. of) the primary antibodies used.

Response 6: We appreciated with this comment. As this comment, we state the clone number, dilution (or conc. of) the primary antibodies used in Supplementary Table 2. These information can be found in Supplementary Table 2.

Material and methods section (new version)

4.9. Preparation of protein and western blotting

To isolate protein from angiotensin â…¡-treated TCMK-1 cells, the cells were scraped using RIPA lysis buffer with phosphatase and proteinase inhibitor (EzRIPA; ATTO; Tokyo, Japan) and incubated on ice for 20 minutes. After centrifuging at 13,000 × g for 20 minutes, at 4 °C, clean supernatants were move to a new tube and the concentration of supernatants was analyzed with bicinchoninic acid assay kit (BCA kit; Thermo Fisher Scientific, Inc.; Waltham, MA, USA). To validate protein expression from TCMK-1 cells, blotting was conducted. Equal amount of lysate proteins (30 μg/lane) was separated by 10% sodium dodecyl sulfate polyacrylamide gel using electrophoresis. Then, running proteins were transferred to polyvinylidene fluoride membranes, which were incubated with diluted primary antibodies (Anti-β-actin, AGTR1, TGF-β, SMAD2/3 and pSMAD2/3) at 4 °C. The incubated membranes with antibodies were thoroughly washed using tris buffered saline with 0.1% Tween 20 and incubated with secondary antibodies for 2 hours at room temperature. All membranes were developed by enhanced chemiluminescence (LAS-4000s; GE Healthcare, Chicago, IL, USA). The antibodies information used in this study can be found in sTable 2.

Supplementary Table 2 (new version)

Antibody name (clone)

Company

Cat. No.

Antibody dilution

β-actin (C4)

Santa cruz Biotechnology

sc-47778

1:500

TGF-β (9016)

Abcam

Ab64715

1:500

pSMAD2/3 (D27F4)

Cell signaling

8828 1:1,000

SMAD2/3 (D7G7)

Cell signaling

8685 1:1,000

AGTR1

Invitrogen

PA5-20812 1:1,000

Point 7: In the conflict of interest, please state whether the interpretation of results in the study has any influences by the sponsoring/commercial partners.

Response 7: We appreciated with this comment. As this comment, we state no any influences by the sponsoring/ commercial partners. Conflict of interest section was changed.

Conflict of interest section (previous version)

The authors declare no conflict of interest.

Conflict of interest section (new version)

The authors declare no competing financial interests by the sponsoring/ commercial partners.

This manuscript is a resubmission of an earlier submission. The following is a list of the peer review reports and author responses from that submission.

Round 1

Reviewer 1 Report

The methodology seems to be ok, the statistics are simple, but for what they set out to do, they didn't really need a more complicated statistic.
The limits part is completely missing.

I see a major negative aspect of their argument. It's a good idea to try to discover new drugs and that's what the Authors are trying to do here. But they use arguments that do not stand that ECE and DK are better than other conventional treatments (ACE inhibitors, angiotensin receptor blockers, renin inhibitors, and aldosterone antagonists) in hypertensive nephropathy.
Quote: "Although several studies have shown that currently available antihypertensive agents (ACE inhibitors, angiotensin receptor blockers, renin inhibitors, and aldosterone antagonists) could ameliorate glomerular injuries, they are much less effective for decreasing tubular interstitial fibrosis (TIF) [28, 29]. "

And they refer to two references from which this is not really understood, in one of the references both glomerular injuries and TIF were reduced and in the second reference, neither glomerular injuries nor TIF were reduced.
So if their motivation was to show that CEE and DK are more effective in treating TIF, then they should have a group of animals treated with conventional hypertensive agents. They don't have that either, and the quoted references don't support that.

"Although synthetic chemical agents such as ACE inhibitors and Ang II receptor blockers were used for the treatment of chronic kidney disease such as hypertensive nephropathy, RAAS blockers might facilitate in the compensatory upregulation of other RAAS components, such as renin [48]." Well, ECE and DK, as they said in the intro, also work as ACE inhibitors. They did not dose renin in their study after treatment with ECE and DK to check if ECE and DK behave better and do not give any reference. And then why force an argument with this idea?!

"Moreover, ACE inhibitors and Ang II receptor blockers only inhibit the activity of these RAAS components rather than decreasing their expression or synthesis. Thus, for the therapy of chronic kidney disease to be more effective, inhibiting the expression of RAAS components is more appropriate than just inhibiting their activity [49]. ECE and DK showed a decreased expression of AT1R / TGFβ / SMAD2 / 3 in the kidney and decreased serum level of Ang II. " Exactly the same problem. They also did not measure the effects of ACE or Ang II receptor blockers to suggest such a comparison. I'm sure there are enough studies in which ACEi and Ang II receptor blockers decrease AT1R / TGFβ / SMAD2 / 3 expression.

So what is new about studies that have already studied the effectiveness of these compounds? It does not specify whether there have been studies of ECE and DK in hypertensive nephropathy. I think it would be better to re-argue, not to make comparisons with conventional agents because one cannot support them, to find out if there have been studies of ECE and DK in hypertensive nephropathy and to try to emphasize honestly. what they bring new to those studies. I'm convinced that this could be an ok article then.

Author Response

Response to Reviewer 1 Comments

The methodology seems to be ok, the statistics are simple, but for what they set out to do, they didn't really need a more complicated statistic.

The limits part is completely missing.

Point 1: I see a major negative aspect of their argument. It's a good idea to try to discover new drugs and that's what the Authors are trying to do here. But they use arguments that do not stand that ECE and DK are better than other conventional treatments (ACE inhibitors, angiotensin receptor blockers, renin inhibitors, and aldosterone antagonists) in hypertensive nephropathy.

Quote: "Although several studies have shown that currently available antihypertensive agents (ACE inhibitors, angiotensin receptor blockers, renin inhibitors, and aldosterone antagonists) could ameliorate glomerular injuries, they are much less effective for decreasing tubular interstitial fibrosis (TIF) [28, 29].

And they refer to two references from which this is not really understood, in one of the references both glomerular injuries and TIF were reduced and in the second reference, neither glomerular injuries nor TIF were reduced.

So if their motivation was to show that CEE and DK are more effective in treating TIF, then they should have a group of animals treated with conventional hypertensive agents. They don't have that either, and the quoted references don't support that.

Response 1: The purpose of our study is not to show the superiority of ECE or DK on decreasing TIF comparing with ACE inhibitors. However, we agreed our statement in the manuscript could bring confusion as reviewer’s comment. The purpose of this study was that ECE or DK could be helpful to attenuate hypertensive nephropathy by decreasing EMT in the kidney. Thus, we changed the manuscript as below;

Discussion section (new version)

3. Discussion

Hypertension is the second leading etiology of ESRD after diabetes [20]. It is hard to predict the severity of hypertensive renal fibrosis with BP, since renal fibrosis could severely progress even when the patient’s BP is not extremely high [20]. Although an-tihypertensive treatments with ACE inhibitors, such as angiotensin receptor blockers, renin inhibitors, and aldosterone antagonists, could reduce the severity of hypertensive kidney disease, they are not enough to prevent the progression of hypertensive nephropathy [21]. Even though the recommended target BP is achieved as below 130/80 mmHg, the treatment strategy for decreasing BP could not delay the progres-sion of hypertensive nephropathy [22]. Although hypertensive nephropathy is typi-cally described as nephroangiosclerosis and glomerular hyalinosis [23, 24], recently, it is revealed that the interstitium of the kidney, which is involved in TIF, is related with disease progression as well as the glomerular and vascular compartments [25, 26]. Since TIF is a main pathophysiology of hypertensive nephropathy, it is essential to the development of new agents directed to modulate TIF to prevent the progression of hy-pertensive nephropathy. In recent years, EMT has been known as a crucial player in renal fibrosis [27, 28].

Point 2: "Although synthetic chemical agents such as ACE inhibitors and Ang II receptor blockers were used for the treatment of chronic kidney disease such as hypertensive nephropathy, RAAS blockers might facilitate in the compensatory upregulation of other RAAS components, such as renin [48]." Well, ECE and DK, as they said in the intro, also work as ACE inhibitors. They did not dose renin in their study after treatment with ECE and DK to check if ECE and DK behave better and do not give any reference. And then why force an argument with this idea?!

Response 2: The purpose of our study was not that reveal the ECE or DK are better treatment comparing with ACE inhibitors by decreasing compensatory upregulation of renin, however it could be confusing as reviewer’s comment by our former description in the manuscript. Thus we changed the description like below;

Discussion section (new version)

3. Discussion

Hypertension tends to increase proteinuria [38]. It is known that the urine albumin-to-creatinine ratio is associated with the decrease in glomerular filtration rate [42]. Previous studies have shown that the development of microalbuminuria in SHRs is caused by predominant tubular injury, which induced the urinary loss of low molecular weight proteins [43, 44]. In our study, the urine albumin-to-creatinine ratio of the SHRs was higher than that of the WKY rats and was decreased by the administration of either ECE or DK.

Previous studies showed that several polyphenols have effect on inhibiting ACE activity [14, 15-17]. However, these studies only reported ACE inhibitory activity as an IC50 value, which is the 50% inhibition concentration of a positive control and they did not show antihypertensive effect or protecting effect against hypertensive nephropathy in the animals. One study showed that ECE showed decreasing BP in the 2-kindney 1-clip Goldblatt hypertensive rats [45]. Pyrogallol-phloroglucinol-6,6-bieckol deceased BP in the high fat diet induced hypertension animal model by modulating vascular dysfunction [46]. However, these studies showed only decreasing BP and did not evaluate whether ECE showed effect of attenuating hypertensive nephropathy. Our study showed ECE and DK decreased expression of AT1R/TGFβ/SMAD2/3 in the kidney and decreased serum level of Ang II. Additionally, ECE and DK decreased EMT and renal fibrosis and decreased glomerular sclerosis. It seemed that ECE and DK might be helpful to attenuate hypertensive nephropathy.

Point 4: So what is new about studies that have already studied the effectiveness of these compounds? It does not specify whether there have been studies of ECE and DK in hypertensive nephropathy. I think it would be better to re-argue, not to make comparisons with conventional agents because one cannot support them, to find out if there have been studies of ECE and DK in hypertensive nephropathy and to try to emphasize honestly. what they bring new to those studies. I'm convinced that this could be an ok article then.

Response 4: As same as the response to reviewer’s comment 3, our study purpose was showed the ECE or DK have attenuating effect on hypertensive nephropathy by decreasing EMT which was not previously revealed. Thus, we rewrote the manuscript like below;

Discussion section (new version)

3. Discussion

Hypertension tends to increase proteinuria [38]. It is known that the urine albumin-to-creatinine ratio is associated with the decrease in glomerular filtration rate [42]. Previous studies have shown that the development of microalbuminuria in SHRs is caused by predominant tubular injury, which induced the urinary loss of low molecular weight proteins [43, 44]. In our study, the urine albumin-to-creatinine ratio of the SHRs was higher than that of the WKY rats and was decreased by the administration of either ECE or DK.

Previous studies showed that several polyphenols have effect on inhibiting ACE activity [14, 15-17]. However, these studies only reported ACE inhibitory activity as an IC50 value, which is the 50% inhibition concentration of a positive control and they did not show antihypertensive effect or protecting effect against hypertensive nephropathy in the animals. One study showed that ECE showed decreasing BP in the 2-kindney 1-clip Goldblatt hypertensive rats [45]. Pyrogallol-phloroglucinol-6,6-bieckol deceased BP in the high fat diet induced hypertension animal model by modulating vascular dysfunction [46]. However, these studies showed only decreasing BP and did not evaluate whether ECE showed effect of attenuating hypertensive nephropathy. Our study showed ECE and DK decreased expression of AT1R/TGFβ/SMAD2/3 in the kidney and decreased serum level of Ang II. Additionally, ECE and DK decreased EMT and renal fibrosis and decreased glomerular sclerosis. It seemed that ECE and DK might be helpful to attenuate hypertensive nephropathy.

Our study showed ECE and DK decreased expression of AT1R/TGFβ/SMAD2/3 in the kidney and decreased serum level of Ang II. Additionally, ECE and DK decreased EMT and renal fibrosis and decreased glomerular sclerosis. It seemed that ECE and DK might be helpful to attenuate hypertensive nephropathy.

In animal experiments, studies using AT1 receptor antagonist were not conducted, but as your opinion, an in vitro model was constructed with angiotensin â…¡-induced renal tubular cells (TCMK-1 cells) to confirm the effects of DK and ECE administration. When 1 μmol/L telmisartan was used as an AT1 receptor antagonist and the changes in TGFβ, SMAD2, and 3, which are important pathways of EMT, were confirmed, DK and ECE showed potential as AT1 receptor antagonists (supporting figure 1). And, if there is a chance in the future, we would like to conduct an animal experiment using the AT1 receptor antagonist.

supporting figure 1

Reviewer 2 Report

COMMENTS FOR THE AUTHOR:

In the present study, authors investigated the renoprotective role of Ecklonia cava extracts and dieckol by decreasing endothelial mesenchymal transition in hypertensive nephropathy. The current study showed that Ecklonia cava extracts and dieckol ameliorates hypertensive nephropathy through the decreased levels of angiotensin II and the inhibition of AT1R/TGFβ/SMAD pathway in spontaneously hypertensive rats. The concepts and many of the findings in the text are interesting. However, this article includes several problems which need to be explained or be rewritten. Respective comments are as follows;

Major Comments

  1. Authors presented that the treatment of Ecklonia cava extracts and dieckol improved renal function in hypertensive nephropathy. However, you did not suggest serum creatinine level, a representative indicator of renal function. Please show the levels of serum creatinine among the experimental groups according to treatment of Ecklonia cava extracts and dieckol in spontaneously hypertensive rats.
  2. Authors suggested that Ecklonia cava extracts and dieckol ameliorates hypertensive nephropathy through the decreased levels of angiotensin II and the inhibition of AT1R/TGFβ/SMAD pathway. However, you did not present whether Ecklonia cava extracts and dieckol directly inhibit AT1R/TGFβ/SMAD pathway. To solve this question, please add the experimental group for the co-treatment of Ang II and Ecklonia cava extracts or dieckol in spontaneously hypertensive rats, or perform in vitro study for the AT1R/TGFβ/SMAD pathway in the co-treatment of Ang II and Ecklonia cava extracts or dieckol in proximal tubular cells.
  3. To definitively confirm molecular signaling pathway for the renoprotective effects of Ecklonia cava extracts and dieckol in hypertensive nephropathy, please show the representative images and semiquantitative analysis for the immunoblot for AT1R/TGFβ/SMAD2/SMAD3/snail2 and vimentin/α-SMA expressions as well as the levels of mRNA in spontaneously hypertensive rats.
  4. In figure 2, 4, and 5, the resolutions and qualities of representative images are very low, and the representative figures are not matched with the graph of semi-quantitative analysis. In addition, please increase the magnification power of light microscope and present the number of total magnification power in each figure.
  5. The dose of Ecklonia cava extracts seems to be relatively high. Is Ecklonia cava extracts clinically safe in high dose setting? Please show the mortality rates of this study and the levels of liver enzymes to confirm hepatotoxicity of these drugs.
  6. How did you decide the dose of dieckol? What is the dose of dieckol 2.5mg/kg equivalent to Ecklonia cava extracts?
  7. The experimental method of urine microalbumin and UACR was not described in the method section. Which method, 24-h urine or spot urine, did you perform to measure urine microalbumin and UACR?

Author Response

Response to Reviewer 2 Comments

In the present study, authors investigated the renoprotective role of Ecklonia cava extracts and dieckol by decreasing endothelial mesenchymal transition in hypertensive nephropathy. The current study showed that Ecklonia cava extracts and dieckol ameliorates hypertensive nephropathy through the decreased levels of angiotensin II and the inhibition of AT1R/TGFβ/SMAD pathway in spontaneously hypertensive rats. The concepts and many of the findings in the text are interesting. However, this article includes several problems which need to be explained or be rewritten. Respective comments are as follows;

Major Comments

Point 1: Authors presented that the treatment of Ecklonia cava extracts and dieckol improved renal function in hypertensive nephropathy. However, you did not suggest serum creatinine level, a representative indicator of renal function. Please show the levels of serum creatinine among the experimental groups according to treatment of Ecklonia cava extracts and dieckol in spontaneously hypertensive rats

Response 1: We appreciated with this comment. As your comment, we want to validate improved renal function in hypertensive nephropathy, and we added creatinine level in urine of spontaneously hypertensive rats (SHR, supporting fig 1). In supporting fig 1, creatinine level increased in SHR but those levels of Ecklonia cava extracts and dieckol administrative SHR statistically decreased. This result added in figure 6 of Manuscript.

Figure 6 and its legend (new version)

Figure 6. Comparative analysis of ECE and DK administration on the attenuation of renal function aggravation in SHRs. (a) water consumption, (b) urine volume, (c) urine Na/K ratio, (d) urine microalbumin level, (e) urine albumin-to-creatinine ratio (UACR) and (f) urine creatine levels were measured prior to sacrifice. Three doses of ECE (50 mg/kg/day, 100 mg/kg/day and 150 mg/kg/day) were oral administrated for 4 weeks and 2.5 mg/kg/day DK also oral administrated for 4 weeks. Means denoted by a different letter indicate significant differences between groups (p < 0.05). ECE, Ecklonia cava extract; DK, dieckol.

Results (new version)

2.6. ECE and DK attenuated renal function aggravation in SHRs

The amount of water intake within 24 hours was not significantly different among all groups (Figure 6a). The urine volume of the SHRs within 24 hours was significantly lower than that of the WKY rats (Figure 6b). It was significantly increased by the administration of 100 and 150 mg/kg/day of ECE and DK. The urine sodium/potassium (Na/K) ratio was not significantly different among all groups (Figure 6c). The albumin level in the urine of the SHRs was significantly higher than that of the WKY rats and was significantly decreased by the administration of ECE or DK. The decreasing effects of 50, 100, and 150 mg/kg/day of ECE and DK were not significantly different (Figure 6d). The ratio of urine albumin/creatinine and urine creatine levels of the SHRs was significantly higher than that of the WKY rats and was decreased by the administration of ECE or DK. The decreasing effects of 50 and 100 mg/kg/day of ECE and DK were not significantly different (Figure 6e and f).

Point 2: Authors suggested that Ecklonia cava extracts and dieckol ameliorates hypertensive nephropathy through the decreased levels of angiotensin II and the inhibition of AT1R/TGFβ/SMAD pathway. However, you did not present whether Ecklonia cava extracts and dieckol directly inhibit AT1R/TGFβ/SMAD pathway. To solve this question, please add the experimental group for the co-treatment of Ang II and Ecklonia cava extracts or dieckol in spontaneously hypertensive rats, or perform in vitro study for the AT1R/TGFβ/SMAD pathway in the co-treatment of Ang II and Ecklonia cava extracts or dieckol in proximal tubular cells.

Response 2: We appreciated with this comment. As your comment, we created an in vitro model using TCMK-1 cells, which is mouse kidney tubular cells, and were produced by treatment with 100 nmol/L angiotensin â…¡ [Reference: Acta Pharmacologica Sinica (2015) 36: 821–830]. After ECE and DK treatment, it was confirmed that ECE and DK suppress angiotensin â…¡-induced EMT signaling including AT1R/TGFβ/SMAD pathway. These results added in Figure 2 and Figure 3 and Figure and Results section were also changed.

Figure 2 and its legend (new version)

Figure 2. Comparative analysis of ECE and DK administration on the reduction of expression of AT1R in the kidney of SHRs and in angiotensin â…¡-treated TCMK-1 cells. (a) AT1R mRNA levels in cortex area, and in medulla of kidney were measured by qRT-PCR analysis. (b) AT1R protein expression levels in cortex area, and in medulla of kidney were measured by immunohistochemistry and (c) the protein levels were quantified by Image J software. (d) AT1R protein expression levels in angiotensin â…¡-treated TCMK-1 cells with 5, 25, 50 ug/ml ECE, 2.5 ug/ml DK or inhibitor were measured by Western blotting. Scale bar = 50 μm. Three doses of ECE (50 mg/kg/day, 100 mg/kg/day and 150 mg/kg/day) was oral administrated for 4 weeks and 2.5 mg/kg/day DK also oral administrated for 4 weeks. Means denoted by a different letter indicate significant differences between groups (p < 0.05). ECE, Ecklonia cava extract; DK, dieckol; Inhibitor, 1 μmol/L telmisartan.

Figure 3 and its legend (new version)

Figure 3. Comparative analysis of ECE and DK administration on the reduction of expression of TGFβ, SMAD2/3, and Snail2 in the kidney of SHRs. (a) TGFβ, (b) SMAD2, (c) SMAD3, (d) Snail2 mRNA levels in cortex area, and in medulla of kidney were measured by qRT-PCR analysis. Three doses of ECE (50 mg/kg/day, 100 mg/kg/day and 150 mg/kg/day) was oral administrated for 4 weeks and 2.5 mg/kg/day DK also oral administrated for 4 weeks. (e) TGFβ, SMAD2/3, pSMAD2/3 protein expression levels in angiotensin â…¡-treated TCMK-1 cells with 5, 25, 50 ug/ml ECE, 2.5 ug/ml DK or inhibitor were measured by Western blotting. Means denoted by a different letter indicate significant differences between groups (p < 0.05). ECE, Ecklonia cava extract; DK, dieckol, Inhibitor, 1 μmol/L telmisartan.

Results (previous version)

2.2. ECE and DK attenuated the expression of AT1R in the kidney of SHRs

The expression of AT1R was evaluated through qRT-PCR and staining. The mRNA expression of AT1R in the medulla and cortex of the kidney was significantly higher in the SHRs than that of the WKY rats. Such expression was decreased by the administration of either ECE or DK (Figure 2a). In the medulla, the decreasing effect was most prominent in the group treated with DK. In the cortex, 150 mg/kg/day of ECE had the most prominent decreasing effect. The expression level of AT1R in the cortex and medulla of the SHRs, which was evaluated through staining, was significantly increased and was significantly decreased by the administration of either ECE or DK. Moreover, the most prominent decreasing effect was observed in the group treated with DK (Figure 2b and c).

2.3. ECE and DK reduced the expression of TGFβ, SMAD2/3, and Snail2 in the kidney of SHRs

The expressions of TGFβ in the medulla and cortex of the SHRs were significantly higher than those of the WKY rats, which were decreased by the administration of ECE or DK. In the medulla, the decreasing effect was most prominent in the groups treated with 150 mg/kg/day of ECE and DK. In the cortex, the decreasing effect was most prominent in the group treated with DK (Figure 3a). The expressions of SMAD2 in the medulla and cortex of the SHRs were higher than those of the WKY rats, which were significantly decreased by the administration of ECE or DK. The decreasing effect in the medulla was most prominent in the groups treated with 150 mg/kg/day of ECE or DK. In the cortex, the decreasing effects of 50, 100, and 150 mg/kg/day of ECE or DK were not significantly different (Figure 3b). The expression of SMAD3 in the medulla and cortex of the SHRs was significantly higher than that of the WKY rats and was significantly decreased by the administration of ECE or DK. The most prominent decreasing effect was observed in the group treated with 150 mg/kg/day of ECE (Figure 3c). The expression of Snail2 in the medulla and cortex of the SHRs was significantly increased by the administration of ECE or DK. The most prominent decreasing effect was observed in the group treated with DK (Figure 3d).

Results (new version)

2.2. ECE and DK attenuated the expression of AT1R in the kidney of SHRs

The expression of AT1R was evaluated through qRT-PCR and staining. The mRNA expression of AT1R in the medulla and cortex of the kidney was significantly higher in the SHRs than that of the WKY rats. Such expression was decreased by the administration of either ECE or DK (Figure 2a). In the medulla, the decreasing effect was most prominent in the group treated with DK. In the cortex, 150 mg/kg/day of ECE had the most prominent decreasing effect. The expression level of AT1R in the cortex and medulla of the SHRs, which was evaluated through staining, was significantly increased and was significantly decreased by the administration of either ECE or DK. Moreover, the most prominent decreasing effect was observed in the group treated with DK (Figure 2b and c). In in vitro model, the expression level of AT1R protein in angiotensin â…¡-treated TCMK-1 cells with ECE (5, 25, 50 ug/ml), DK or inhibitor was decreased than only angiotensin â…¡ treatment (Figure 2d).

2.3. ECE and DK reduced the expression of TGFβ, SMAD2/3, and Snail2 in the kidney of SHRs

The expressions of TGFβ in the medulla and cortex of the SHRs were significantly higher than those of the WKY rats, which were decreased by the administration of ECE or DK. In the medulla, the decreasing effect was most prominent in the groups treated with 150 mg/kg/day of ECE and DK. In the cortex, the decreasing effect was most prominent in the group treated with DK (Figure 3a). The expressions of SMAD2 in the medulla and cortex of the SHRs were higher than those of the WKY rats, which were significantly decreased by the administration of ECE or DK. The decreasing effect in the medulla was most prominent in the groups treated with 150 mg/kg/day of ECE or DK. In the cortex, the decreasing effects of 50, 100, and 150 mg/kg/day of ECE or DK were not significantly different (Figure 3b). The expression of SMAD3 in the medulla and cortex of the SHRs was significantly higher than that of the WKY rats and was significantly decreased by the administration of ECE or DK. The most prominent decreasing effect was observed in the group treated with 150 mg/kg/day of ECE (Figure 3c). The expression of Snail2 in the medulla and cortex of the SHRs was significantly increased by the administration of ECE or DK. The most prominent decreasing effect was observed in the group treated with DK (Figure 3d). In in vitro model, the expression level of TGFβ and pSMAD2/3 protein in angiotensin â…¡-treated TCMK-1 cells with ECE (5, 25, 50 ug/ml), DK or inhibitor was decreased than only angiotensin â…¡ treatment (Figure 3e).

Material and methods (new version)

4.8. In vitro modeling using TCMK-1 cells

TCMK-1 cells, a mouse proximal tubule cell line, were purchased from American Type Culture Collection (Washington, DC, USA). High glucose Dulbecco's Modified Eagle's medium (Gibco; Grand island, NY, USA), 10% fetal bovine serum (FBS) and 1% penicillin-streptomycin were used as culture medium. To validate the inhibitory effects of ECE and DK in angiotensin â…¡-treated TCMK-1 cells, we treated ECE (5, 25, 50 μg/mL) and DK (1.8 μg/mL) with 100 nmol/L angiotensin â…¡ (Sigma-Aldrich, St Louis, MO, USA) for 12 hours. Telmisartan (1 μmol/L, Sigma-Aldrich) was used for AT1 receptor antagonist and pretreated before angiotensin â…¡ for 3 hours

4.9. Preparation of protein and western blotting

To isolate protein from angiotensin â…¡-treated TCMK-1 cells, the cells were scraped using RIPA lysis buffer with phosphatase and proteinase inhibitor (EzRIPA; ATTO; Tokyo, Japan) and incubated on ice for 20 minutes. After centrifuging at 13,000 × g for 20 minutes, at 4 °C, clean supernatants were move to a new tube and the concentration of supernatants was analyzed with bicinchoninic acid assay kit (BCA kit; Thermo Fisher Scientific, Inc.; Waltham, MA, USA). To validate protein expression from TCMK-1 cells, blotting was conducted. Equal amount of lysate proteins (30 μg/lane) was separated by 10% sodium dodecyl sulfate polyacrylamide gel using electrophoresis. Then, running proteins were transferred to polyvinylidene fluoride membranes, which were incubated with diluted primary antibodies (Anti-β-actin, AGTR1, TGF-β, SMAD2/3 and pSMAD2/3) at 4 °C. The incubated membranes with antibodies were thoroughly washed using tris buffered saline with 0.1% Tween 20 and incubated with secondary antibodies for 2 hours at room temperature. All membranes were developed by enhanced chemiluminescence (LAS-4000s; GE Healthcare, Chicago, IL, USA).

Point 3: To definitively confirm molecular signaling pathway for the renoprotective effects of Ecklonia cava extracts and dieckol in hypertensive nephropathy, please show the representative images and semiquantitative analysis for the immunoblot for AT1R/TGFβ/SMAD2/SMAD3/snail2 and vimentin/α-SMA expressions as well as the levels of mRNA in spontaneously hypertensive rats.

Response 3: We appreciated with this comment. To prevent contamination with other type cells and to check the results in cortex layer or medullar layer separately, we performed immunohistochemistry for validating EMT markers including vimentin/α-SMA and AT1R of SHR rat kidney sample. Molecules related to pathways including TGFβ/SMAD2/SMAD3 were only confirmed at the mRNA level, but next time we have the opportunity, we will check the protein levels in SHR rat kidney.

Point 4: In figure 2, 4, and 5, the resolutions and qualities of representative images are very low, and the representative figures are not matched with the graph of semi-quantitative analysis. In addition, please increase the magnification power of light microscope and present the number of total magnification power in each figure.

Response 4: We appreciated with this comment. As of your opinion, we have added a high-magnification image to all the images (x200 à x400). The changed image can be seen in figures 2, 4, and 5.

 Figure 2 (new version)

Figure 4 (new version)

Figure 5 (new version)

Point 5: The dose of Ecklonia cava extracts seems to be relatively high. Is Ecklonia cava extracts clinically safe in high dose setting? Please show the mortality rates of this study and the levels of liver enzymes to confirm hepatotoxicity of these drugs.

Response 5: We appreciated with this comment. Polyphenol extract from Ecklonia cava are clinically safe in 144 mg/day dosage. In clinical study, effects of 12-week supplementation with a polyphenol extract from Ecklonia cava (ECP) on anthropometry, serum biochemistry and hematology have been investigated. Ninety-seven overweight male and female adults (average age 40.5 ± 9.2 yr and body mass index (BMI) of 26.5 ± 1.6 kg/m²) were enrolled in a randomized, double-blind, placebo-controlled trial with parallel-group design. Subjects were randomly allocated into three groups designated as placebo, low-dose (72 mg-ECP/day) and high-dose (144 mg-ECP/day). In this study, results demonstrated that ECP supplementation significantly contributed to lowering body fat and serum lipid parameters such as total and LDL cholesterols with dose dependence. Further studies using different populations, dosages or biological markers are highly recommended to better understand the physiological features of this polyphenol. [Reference: Phytother Res. 2012 Mar;26(3):363-8.]

Point 6: How did you decide the dose of dieckol? What is the dose of dieckol 2.5mg/kg equivalent to Ecklonia cava extracts?

Response 6: In our previous study, to confirm the co-elution of two nearby peaks, we performed several HPLC and put a representative image (supporting figure 1A). Separated each compound to confirm purity. The separated compounds show a single peak on HPLC (supporting figure 1B) and the purity of each compound shows that DK is 93.58 %, PHB is 92.35 %, PPB is 91.24 % and PFFA is 94.78 % (supporting figure 1C). DK is contained in a 2% content of Ecklonia cava extract as below [Reference: Mar Drugs. 2018,16, 441.; Supporting figure 1]; Therefore, DK contained in 150 mg/kg ECE is 3 mg/kg. However, through the results of this study, we confirmed that ECE was generally effective at 100 mg/kg ECE, and there were factors that had better efficacy at 150 mg/kg ECE. Accordingly, the efficacy was evaluated using 2.5 mg/kg DK, an intermediate content of 2 mg/kg DK contained in 100 mg/kg ECE and 3 mg/kg DK contained in 150 mg/kg ECE.

 Supporting figure 1 [Ref. Supplementary Figure 1. Mar Drugs. 2018,16, 441.]

Supplementary Figure 1. HPLC chromatograms and purity of four compounds from E. cava extract

(A) HPLC chromatograms of E. cava. The peaks labeled a-d correspond to dieckol, PHB, PPB and PFFA, respectively and checked purity of them. a: dieckol, b: 2,7-phloroglucinol-6,6-bieckol (PHB), c: pyrogallol-phloroglucinol-6,6-bieckol (PPB), d: phlorofucofuroeckol-A (PFFA). (B) Separately isolated single compound was validated HPLC chromatograms. (C) Purity of isolated four compounds from E. cava extract

Point 7: The experimental method of urine microalbumin and UACR was not described in the method section. Which method, 24-h urine or spot urine, did you perform to measure urine microalbumin and UACR?

Response 7: We appreciated with this comment. As your comment, experimental method of urine microalbumin and UACR added in material and methods section. To measure urine microalbumin and UACR, the rats could stay in the metabolic cage for 24 hours to collect urine from the rats, and then the experiment was conducted (supporting image 1).

Supporting image 1 and information

This metabolic cage designed specifically for mice features a unique funnel and cone design that effectively separates feces and urine into tubes outside the cage. This cage is designed to allow for immediate, complete, and total separation of urine and feces.

The cage has a two-part feeding chamber located outside of the cage itself. The front chamber catches spilled food so there is no possibility of food/waste contamination. The water bottle has a special drain to collect overflow while the animal is drinking and diverts it into a collection tube separate from the waste collection tubes. The feeding drawers slide out easily, allowing for removal without interrupting the study.

Material and methods (previous version)

4.2. Hypertension animal model

Male SHRs (8 weeks old) and male WKY rats (8 weeks old) were bought from the Orient Bio (Sungnam, Korea) and kept at a constant temperature of approximately 24°C, relative humidity of 50%–55%, and light/dark cycle of 12 hours/12 hours. The rat sublimation was conducted for 1 week. The rats were randomly divided into six groups (five rats per group):

(i) WKY group, in which the rats were orally administered with drinking water for 4 weeks;

(ii) SHR group, in which the rats were orally administered with drinking water for 4 weeks;

(iii–v) SHR group, in which the rats were orally administered with ECE of (iii) 50 mg/kg/day, (iv) 100 mg/kg/day, and (v) 150 mg/kg/day for 4 weeks;

(vi) SHR group, in which the rats were orally administered with 2.5 mg/kg/day DK for 4 weeks. After 4 weeks, all rats were sacrificed according to the ethical principles of the Institutional Animal Care and Use Committee of Gachon University (approval number: LCDI-2019-0121).

Material and methods (previous version)

4.2. Hypertension animal model

Male SHRs (8 weeks old) and male WKY rats (8 weeks old) were bought from the Orient Bio (Sungnam, Korea) and kept at a constant temperature of approximately 24°C, relative humidity of 50%–55%, and light/dark cycle of 12 hours/12 hours. The rat sublimation was conducted for 1 week. The rats were randomly divided into six groups (five rats per group):

(i) WKY group, in which the rats were orally administered with drinking water for 4 weeks;

(ii) SHR group, in which the rats were orally administered with drinking water for 4 weeks;

(iii–v) SHR group, in which the rats were orally administered with ECE of (iii) 50 mg/kg/day, (iv) 100 mg/kg/day, and (v) 150 mg/kg/day for 4 weeks;

(vi) SHR group, in which the rats were orally administered with 2.5 mg/kg/day DK for 4 weeks. After 4 weeks, water consumption, urine volume and urine chemical analysis using by metabolic cage for 24 hours. After completing metabolic analysis, all rats were sacrificed according to the ethical principles of the Institutional Animal Care and Use Committee of Gachon University (approval number: LCDI-2019-0121).

Reviewer 3 Report

The authors should consider the followings:

  1. Across the experiments, please give rationales why no positive controls were given in parallel to the study.
  2. Please give rationale(s), of why the ranges of dose of ECE and DK, were selected to the study.
  3. Please give rationales why the particular "housekeeping" gene in the qPCR was selected; please provide evidences to support that the gene is of housekeeping purposes to the study theme.
  4. Any stability study performed to validate the stability of ECE under various storage terms?
  5. The author should state the % yield of DK from ECE. A chemical profile (such as by HPLC or LCMS analysis) should be provided to give a considerable information to ECE and DK.
  6. The authors should state clearly the novelty of this research in your abstract and conclusion.
  7. Since AT1R/ TGB/ SMAD were intended to study in this study, please explain why the authors not using Knockout or Knockin mice/rats for this investigation?
  8. The authors should explain why they chose a particular route of administration and the doses used in the study.
  9. Please provide details how ECE was authenticated (by pharmacopoeia, chemical marker? means of analyses? and by botantist?)
  10. Did the batch of ECE (extract) being well-stored for traceability purposes?
  11. The authors may consider using English proof-reading services by language professional.
  12. The authors should declare whether they have any conflicting interests.
  13. The authors should consider running protein assay, such as western blotting, for quantitative anaylses of the interested protein.
  14. Please provide the limit of detection to the ELISA assay.
  15. A graphical abstract or figure of summary may be provided as a figure for a wider group of audience.

Author Response

Response to Reviewer 3 Comments

Point 1: Across the experiments, please give rationales why no positive controls were given in parallel to the study.

Response 1: We appreciate this comment. As your comment, we created an in vitro model using TCMK-1 cells, which is mouse kidney tubular cells, and were produced by treatment with 100 nmol/L angiotensin â…¡ [Reference: Acta Pharmacologica Sinica (2015) 36: 821–830]. Positive control (Telmisartan, AT1 receptor antagonist) was used. After ECE and DK treatment, it was confirmed that ECE and DK suppress angiotensin â…¡-induced EMT signaling including AT1R/TGFβ/SMAD pathway. These results added in Figure 2 and Figure 3 and Figure and Results section were also changed.

Figure 2 and its legend (new version)

Figure 2. Comparative analysis of ECE and DK administration on the reduction of expression of AT1R in the kidney of SHRs and in angiotensin â…¡-treated TCMK-1 cells. (a) AT1R mRNA levels in cortex area, and in medulla of kidney were measured by qRT-PCR analysis. (b) AT1R protein expression levels in cortex area, and in medulla of kidney were measured by immunohistochemistry and (c) the protein levels were quantified by Image J software. (d) AT1R protein expression levels in angiotensin â…¡-treated TCMK-1 cells with 5, 25, 50 ug/ml ECE, 2.5 ug/ml DK or inhibitor were measured by Western blotting. Scale bar = 50 μm. Three doses of ECE (50 mg/kg/day, 100 mg/kg/day and 150 mg/kg/day) was oral administrated for 4 weeks and 2.5 mg/kg/day DK also oral administrated for 4 weeks. Means denoted by a different letter indicate significant differences between groups (p < 0.05). ECE, Ecklonia cava extract; DK, dieckol; Inhibitor, 1 μmol/L telmisartan.

Figure 3 and its legend (new version)

Figure 3. Comparative analysis of ECE and DK administration on the reduction of expression of TGFβ, SMAD2/3, and Snail2 in the kidney of SHRs. (a) TGFβ, (b) SMAD2, (c) SMAD3, (d) Snail2 mRNA levels in cortex area, and in medulla of kidney were measured by qRT-PCR analysis. Three doses of ECE (50 mg/kg/day, 100 mg/kg/day and 150 mg/kg/day) was oral administrated for 4 weeks and 2.5 mg/kg/day DK also oral administrated for 4 weeks. (e) TGFβ, SMAD2/3, pSMAD2/3 protein expression levels in angiotensin â…¡-treated TCMK-1 cells with 5, 25, 50 ug/ml ECE, 2.5 ug/ml DK or inhibitor were measured by Western blotting. Means denoted by a different letter indicate significant differences between groups (p < 0.05). ECE, Ecklonia cava extract; DK, dieckol, Inhibitor, 1 μmol/L telmisartan.

Results (previous version)

2.2. ECE and DK attenuated the expression of AT1R in the kidney of SHRs

The expression of AT1R was evaluated through qRT-PCR and staining. The mRNA expression of AT1R in the medulla and cortex of the kidney was significantly higher in the SHRs than that of the WKY rats. Such expression was decreased by the administration of either ECE or DK (Figure 2a). In the medulla, the decreasing effect was most prominent in the group treated with DK. In the cortex, 150 mg/kg/day of ECE had the most prominent decreasing effect. The expression level of AT1R in the cortex and medulla of the SHRs, which was evaluated through staining, was significantly increased and was significantly decreased by the administration of either ECE or DK. Moreover, the most prominent decreasing effect was observed in the group treated with DK (Figure 2b and c).

2.3. ECE and DK reduced the expression of TGFβ, SMAD2/3, and Snail2 in the kidney of SHRs

The expressions of TGFβ in the medulla and cortex of the SHRs were significantly higher than those of the WKY rats, which were decreased by the administration of ECE or DK. In the medulla, the decreasing effect was most prominent in the groups treated with 150 mg/kg/day of ECE and DK. In the cortex, the decreasing effect was most prominent in the group treated with DK (Figure 3a). The expressions of SMAD2 in the medulla and cortex of the SHRs were higher than those of the WKY rats, which were significantly decreased by the administration of ECE or DK. The decreasing effect in the medulla was most prominent in the groups treated with 150 mg/kg/day of ECE or DK. In the cortex, the decreasing effects of 50, 100, and 150 mg/kg/day of ECE or DK were not significantly different (Figure 3b). The expression of SMAD3 in the medulla and cortex of the SHRs was significantly higher than that of the WKY rats and was significantly decreased by the administration of ECE or DK. The most prominent decreasing effect was observed in the group treated with 150 mg/kg/day of ECE (Figure 3c). The expression of Snail2 in the medulla and cortex of the SHRs was significantly increased by the administration of ECE or DK. The most prominent decreasing effect was observed in the group treated with DK (Figure 3d).

Results (new version)

2.2. ECE and DK attenuated the expression of AT1R in the kidney of SHRs

The expression of AT1R was evaluated through qRT-PCR and staining. The mRNA expression of AT1R in the medulla and cortex of the kidney was significantly higher in the SHRs than that of the WKY rats. Such expression was decreased by the administration of either ECE or DK (Figure 2a). In the medulla, the decreasing effect was most prominent in the group treated with DK. In the cortex, 150 mg/kg/day of ECE had the most prominent decreasing effect. The expression level of AT1R in the cortex and medulla of the SHRs, which was evaluated through staining, was significantly increased and was significantly decreased by the administration of either ECE or DK. Moreover, the most prominent decreasing effect was observed in the group treated with DK (Figure 2b and c). In in vitro model, the expression level of AT1R protein in angiotensin â…¡-treated TCMK-1 cells with ECE (5, 25, 50 ug/ml), DK or inhibitor was decreased than only angiotensin â…¡ treatment (Figure 2d).

2.3. ECE and DK reduced the expression of TGFβ, SMAD2/3, and Snail2 in the kidney of SHRs

The expressions of TGFβ in the medulla and cortex of the SHRs were significantly higher than those of the WKY rats, which were decreased by the administration of ECE or DK. In the medulla, the decreasing effect was most prominent in the groups treated with 150 mg/kg/day of ECE and DK. In the cortex, the decreasing effect was most prominent in the group treated with DK (Figure 3a). The expressions of SMAD2 in the medulla and cortex of the SHRs were higher than those of the WKY rats, which were significantly decreased by the administration of ECE or DK. The decreasing effect in the medulla was most prominent in the groups treated with 150 mg/kg/day of ECE or DK. In the cortex, the decreasing effects of 50, 100, and 150 mg/kg/day of ECE or DK were not significantly different (Figure 3b). The expression of SMAD3 in the medulla and cortex of the SHRs was significantly higher than that of the WKY rats and was significantly decreased by the administration of ECE or DK. The most prominent decreasing effect was observed in the group treated with 150 mg/kg/day of ECE (Figure 3c). The expression of Snail2 in the medulla and cortex of the SHRs was significantly increased by the administration of ECE or DK. The most prominent decreasing effect was observed in the group treated with DK (Figure 3d). In in vitro model, the expression level of TGFβ and pSMAD2/3 protein in angiotensin â…¡-treated TCMK-1 cells with ECE (5, 25, 50 ug/ml), DK or inhibitor was decreased than only angiotensin â…¡ treatment (Figure 3e).

Material and methods (new version)

4.8. In vitro modeling using TCMK-1 cells

TCMK-1 cells, a mouse proximal tubule cell line, were purchased from American Type Culture Collection (Washington, DC, USA). High glucose Dulbecco's Modified Eagle's medium (Gibco; Grand island, NY, USA), 10% fetal bovine serum (FBS) and 1% penicillin-streptomycin were used as culture medium. To validate the inhibitory effects of ECE and DK in angiotensin â…¡-treated TCMK-1 cells, we treated ECE (5, 25, 50 μg/mL) and DK (1.8 μg/mL) with 100 nmol/L angiotensin â…¡ (Sigma-Aldrich, St Louis, MO, USA) for 12 hours. Telmisartan (1 μmol/L, Sigma-Aldrich) was used for AT1 receptor antagonist and pretreated before angiotensin â…¡ for 3 hours

4.9. Preparation of protein and western blotting

To isolate protein from angiotensin â…¡-treated TCMK-1 cells, the cells were scraped using RIPA lysis buffer with phosphatase and proteinase inhibitor (EzRIPA; ATTO; Tokyo, Japan) and incubated on ice for 20 minutes. After centrifuging at 13,000 × g for 20 minutes, at 4 °C, clean supernatants were move to a new tube and the concentration of supernatants was analyzed with bicinchoninic acid assay kit (BCA kit; Thermo Fisher Scientific, Inc.; Waltham, MA, USA). To validate protein expression from TCMK-1 cells, blotting was conducted. Equal amount of lysate proteins (30 μg/lane) was separated by 10% sodium dodecyl sulfate polyacrylamide gel using electrophoresis. Then, running proteins were transferred to polyvinylidene fluoride membranes, which were incubated with diluted primary antibodies (Anti-β-actin, AGTR1, TGF-β, SMAD2/3 and pSMAD2/3) at 4 °C. The incubated membranes with antibodies were thoroughly washed using tris buffered saline with 0.1% Tween 20 and incubated with secondary antibodies for 2 hours at room temperature. All membranes were developed by enhanced chemiluminescence (LAS-4000s; GE Healthcare, Chicago, IL, USA).

Point 2: Please give rationale(s), of why the ranges of dose of ECE and DK, were selected to the study.

Response 2: We appreciate this comment. In our previous study, to confirm the co-elution of two nearby peaks, we performed several HPLC and put a representative image (supporting figure 1A). Separated each compound to confirm purity. The separated compounds show a single peak on HPLC (supporting figure 1B) and the purity of each compound shows that DK is 93.58 %, PHB is 92.35 %, PPB is 91.24 % and PFFA is 94.78 % (supporting figure 1C). DK is contained in a 2% content of Ecklonia cava extract as below [Ref. Mar Drugs. 2018,16, 441.; Supporting figure 1]; Therefore, DK contained in 150 mg/kg ECE is 3 mg/kg. However, through the results of this study, we confirmed that ECE was generally effective at 100 mg/kg ECE, and there were factors that had better efficacy at 150 mg/kg ECE. Accordingly, the efficacy was evaluated using 2.5 mg/kg DK, an intermediate content of 2 mg/kg DK contained in 100 mg/kg ECE and 3 mg/kg DK contained in 150 mg/kg ECE.

 Supporting figure 1 [Ref. Supplementary Figure 1. Mar Drugs. 2018,16, 441.]

Supplementary Figure 1. HPLC chromatograms and purity of four compounds from E. cava extract

(A) HPLC chromatograms of E. cava. The peaks labeled a-d correspond to dieckol, PHB, PPB and PFFA, respectively and checked purity of them. a: dieckol, b: 2,7-phloroglucinol-6,6-bieckol (PHB), c: pyrogallol-phloroglucinol-6,6-bieckol (PPB), d: phlorofucofuroeckol-A (PFFA). (B) Separately isolated single compound was validated HPLC chromatograms. (C) Purity of isolated four compounds from E. cava extract

Point 3: Please give rationales why the particular "housekeeping" gene in the qPCR was selected; please provide evidences to support that the gene is of housekeeping purposes to the study theme.

Response 3: We appreciate this comment. According to your opinion, a house keeping gene should be used to proceed with qRT-PCR. We used ACTB (Actin Beta) as a house keeping gene in this experiment. ACTB is a paper used as a house keeping gene, and the following references can be found [Reference: BMC Immunol. 2020 Jan 31;21(1):4.]. The sequence of primers used can be found in the supplementary file.

Supporting table 1

sTable 1. List of primer for quantitative polymerase chain reaction

Gene

Primer

TGFβ

Forward

5'- GAGCCCTGGATACCAACTACTG -3’

Reverse

5'- AACCCAGGTCCTTCCTAAAGTC -3’

SMAD2

Forward

5'- GAACTCGGAGAGGTTCTGCTTA -3’

Reverse

5'- CTCCCCTTCCTATATGCCTTCT -3’

SMAD3

Forward

5'- GGGGCTCTGTACATACCTTGAG -3’

Reverse

5'- AGAAACACTGGCACTCTGACAA -3’

AT1R

Forward

5'- TAGCCAAAGGAAGAGTCAGGAG -3’

Reverse

5'- GGAACATAGCAAAGGGAGACTG -3’

Snail2

Forward

5'- GGCCTTTCTCCTCTTACTGGAT -3’

Reverse

5'- TGTGATCCTTGGATGAAGTGTC -3’

Actb

Forward

5’- ACAAAGCTGTTCAGTGTCTCCA -3’

Reverse

5’- CTCCGTTTCCAGAATACACACA -3’

Point 4: Any stability study performed to validate the stability of ECE under various storage terms?

Response 4: We appreciate this comment. This point is related with point 10. One study showed activity of E. cava acetone fraction (ECAF) as not affected by any significant variation in temperature, pH, or exposure to artificial UV light or natural sunlight. Therefore, it will be a very useful or applicable raw material for goods such as functional foods [Reference: Toxicology and Environmental Health Sciences volume 6, page s61–66(2014)].

Point 5: The author should state the % yield of DK from ECE. A chemical profile (such as by HPLC or LCMS analysis) should be provided to give a considerable information to ECE and DK.

Response 5: We appreciate this comment. In our previous study, we performed several HPLC. The separated DK compounds show a single peak on HPLC (supporting figure 1) and the purity of each compound shows that DK is 93.58 % (supporting figure 1) as below [Reference: Mar Drugs. 2018,16, 441.; Supporting figure 1].

Supporting figure 1

Point 6: The authors should state clearly the novelty of this research in your abstract and conclusion.

Response 6: We appreciate this comment. The novelty of our study was that ECE or DK attenuated hypertensive nephropathy by decreasing EMT and renal fibrosis. Previous studies showed that ECE had inhibitory action of ACE by presented IC50 or showed antihypertensive effect of ECE in hypertensive animal models without showing effect on hypertensive nephropathy. Thus we changed our abstract and discussion like below;

Abstract (new version)

Hypertension induces renal fibrosis or tubular interstitial fibrosis, which eventually results in end-stage renal disease. Epithelial-to-mesenchymal transition (EMT) is one of the underlying mechanisms of renal fibrosis. Though previous studies showed Ecklonia cava extracts (ECE) and dieckol (DK) had inhibitory action on angiotensin (Ang) I-converting enzyme which converse Ang I to Ang II. It is known that Ang II involves in renal fibrosis, however it was not evaluated whether ECE or DK attenuated hypertensive nephropathy by decreasing EMT. In this study, the effect of ECE and DK on decreasing Ang II and its down signal pathway of angiotensin type 1 receptor (AT1R)/TGFβ/SMAD, which is related with the EMT and restoring renal function in spontaneously hypertensive rats (SHRs), was investigated. Either ECE or DK significantly decreased the serum level of Ang II in the SHRs. Moreover, the renal expression of AT1R/TGFβ/SMAD was decreased by the administration of either ECE or DK. The mesenchymal cell markers in the kidney of SHRs was significantly decreased by ECE or DK. The fibrotic tissue of the kidney of SHRs was also significantly decreased by ECE or DK. The ratio of urine albumin/creatinine of SHRs was significantly decreased by ECE or DK. Overall, the results of this study indicate that ECE and DK decreased the serum levels of Ang II and expression of AT1R/TGFβ/SMAD and then decreased the EMT and renal fibrosis in SHRs. Furthermore, the decrease of EMT and renal fibrosis could lead to the restoration of renal function. It seemed that ECE or DK could be beneficial for decreasing hypertensive nephropathy by decreasing EMT and renal fibrosis.

Discussion (new version)

Previous studies showed that several polyphenols have effect on inhibiting ACE activity [14, 15-17]. However, these studies only reported ACE inhibitory activity as an IC50 value, which is the 50% inhibition concentration of a positive control and they did not show antihypertensive effect or protecting effect against hypertensive nephropathy in the animals. One study showed that ECE showed decreasing BP in the 2-kindney 1-clip Goldblatt hypertensive rats [45]. Pyrogallol-phloroglucinol-6,6-bieckol deceased BP in the high fat diet induced hypertension animal model by modulating vascular dysfunction [46]. However, these studies showed only decreasing BP and did not evaluate whether ECE showed effect of attenuating hypertensive nephropathy. Our study showed ECE and DK decreased expression of AT1R/TGFβ/SMAD2/3 in the kidney and decreased serum level of Ang II. Additionally, ECE and DK decreased EMT and renal fibrosis and decreased glomerular sclerosis. It seemed that ECE and DK might be helpful to attenuate hypertensive nephropathy.

Point 7: Since AT1R/ TGB/ SMAD were intended to study in this study, please explain why the authors not using Knock-out or Knock-in mice/rats for this investigation?

Response 7: We appreciate this comment. In this study, we wanted to check whether Ecklonia cava is effective as a functional food, and we tried to verify the mechanism by confirming whether the functional ingredient is DK among phlorotannins from Ecklonia cava. As you said, it is better to use Knock-out or Knock-in animal for the accuracy of the mechanism, so we will proceed in the further study.

Point 8: The authors should explain why they chose a particular route of administration and the doses used in the study.            

Response 8: We appreciate this comment. We wanted to confirm the possibility of blood pressure control through intake of Ecklonia cava extract through this study, so we proceeded with oral administration. As for the solution to be dissolved, the volume of saline was appropriate to about 100 ul per animal, and the concentration was set as a preliminary experiment, and it was confirmed that it is effective from 100 mg/kg/day [Reference: Antioxidants (Basel). 2021 Feb 16;10(2):298].

Point 9: Please provide details how ECE was authenticated (by pharmacopoeia, chemical marker? means of analyses? and by botantist?)

Response 9: We appreciate this comment. Ecklonia cava is a nutrient-rich marine alga found in the shallow waters of Jeju, Korea. It has often been used in traditional medicine. Ecklonia cava may improve circulation, reduce inflammation, and protect against obesity and heart disease — though most of the research was conducted in animals or cells [Reference: Antioxidants (Basel). 2021 Feb 16;10(2):298; Oxid Med Cell Longev. 2021 Jan 26;2021:8869085; Mar Drugs. 2020 Dec 16;18(12):648; Nutrients. 2020 Sep 11;12(9):2777.].

Point 10: Did the batch of ECE (extract) being well-stored for traceability purposes?

Response 10: We appreciate this comment. ECE used in this study is known to be stored at room temperature and can be stored stably without material change for up to 2 years. One study showed activity of E. cava acetone fraction (ECAF) as not affected by any significant variation in temperature, pH, or exposure to artificial UV light or natural sunlight. Therefore, it will be a very useful or applicable raw material for goods such as functional foods [Reference: Toxicology and Environmental Health Sciences volume 6, page s61–66(2014)].

Point 11: The authors may consider using English proof-reading services by language professional.

Response 11: We appreciate this comment. We checked the manuscript for revision thoroughly and corrected few grammatical and spelling errors in the manuscript. In addition, the grammatical errors were minimized by giving proofreading to native speakers and the relevant supporting documents are attached below.

Certificate of editing

Point 12: The authors should declare whether they have any conflicting interests.

Response 12: We appreciated with this comment. we already declared whether we don’t have any conflicting interests in Conflict of interest section of manuscript (line 444, page 12).

Conflicts of Interest: The authors declare no conflict of interest.

Point 13: The authors should consider running protein assay, such as western blotting, for quantitative anaylses of the interested protein.

Response 13: We appreciate this comment. As your opinion, it would be nice to perform western blotting to confirm the expression of the protein, but in this study, we wanted to specifically observe glomerulus and tubular cells in the cortex and medulla regions in kidney tissue.

Point 14: Please provide the limit of detection to the ELISA assay.

Response 14: We appreciate this comment. ELISA (enzyme-linked immunosorbent assay) is a plate-based assay technique designed for detecting and quantifying soluble substances such as peptides, proteins, antibodies, and hormones. Other names, such as enzyme immunoassay (EIA), are also used to describe the same technology. In an ELISA, the antigen (target macromolecule) is immobilized on a solid surface (microplate) and then complexed with an antibody that is linked to a reporter enzyme. Detection is accomplished by measuring the activity of the reporter enzyme via incubation with the appropriate substrate to produce a measurable product. The most crucial element of an ELISA is a highly specific antibody-antigen interaction (supporting figure 1).

Supporting figure 1

Detection range of used ELISA kit in this study (Rat angiotensin II ELISA Kit; Mybiosource, cat. MBS7606574) is 31.25-2000pg/ml and sensitivity is above 18.75pg/ml.

Point 15: A graphical abstract or figure of summary may be provided as a figure for a wider group of audience.

Response 15: We appreciated with this comment. As this comment, we added graphical abstract in manuscript as figure 7.

Figure 7 and its legend (new version)

Figure 7. Schematic image for effects of ECE and DK administration in SHRs. ECE or DK downregulated AT1R/TGFβ/SMAD2 or 3 molecule expression and EMT marker expression in SHRs. 
